# Astronomical aspects of Group E-type complexes and implications for understanding ancient Maya architecture and urban planning

**Ivan Šprajc**⬚*

Institute of Anthropological and Spatial Studies, Research Center of the Slovenian Academy of Sciences and Arts (ZRC SAZU), Ljubljana, Slovenia

* sprajc@zrc-sazu.si

## Abstract

In the 1920s, during the first archaeological excavations at Uaxactún, Petén, Guatemala, an architectural complex named Group E was interpreted as an ancient Maya astronomical observatory, intended specifically for sighting the equinoctial and solstitial sunrises. In the following decades, a large number of architectural compounds with the same configuration have been found, most of them in the central lowlands of the Yucatan peninsula. The multiple hypotheses that have been proposed about the astronomical function of these complexes, commonly designated as E Groups, range from those attributing them a paramount role in astronomical observations to those that consider them merely allegorical or commemorative allusions to celestial cycles, without any observational use. This study, based on quantitative analyses of a reasonably large sample of alignment data, as well as on contextual evidence, shows that many of the previous hypotheses cannot be sustained. I argue that E Groups, although built primarily for ritual purposes, were astronomically functional, but also that they had no specific or particularly prominent role in astronomical observations. Their orientations belong to widespread alignment groups, mostly materialized in buildings of other types and explicable in terms of some fundamental concerns of the agriculturally-based Maya societies. I present the evidence demonstrating that the astronomical orientations initially embedded in E Groups, which represent the earliest standardized form of Maya monumental architecture and whose occurrence in practically all early cities in the central Yucatan peninsula attests to their socio-political significance, were later transferred to buildings and compounds of other types. Therefore, it is precisely the importance of the astronomically and cosmologically significant directions, first incorporated in E Groups, that allows us to understand some prominent aspects of ancient Maya architecture and urbanism.

## Introduction

Ever since the 1920s, when Frans Blom and Oliver Ricketson interpreted Group E of Uaxactún as an equinoctial and solstitial observatory [1, 2], it has been assumed that the architectural

**Data Availability Statement:** Lidar data used in this study cannot be shared publicly because of the need to protect archaeological sites from looting. Digital files obtained by lidar surveys in Belize, the Ceibal area and within the Pacunam Lidar Initiative

in Guatemala are on file with the Institute of Archaeology in Belize (Cor. Culvert Road and Mountain View Blvd, Belmopan City, Belize, ia@nichbelize.org, https://nichbelize.org/institute-of-archaeology/), the Instituto de Antropología e Historia de Guatemala (IDAEH, 12 Ave. 11-11, Zona 1, Guatemala City, vu.demopre@gmail.com; http://mcd.gob.gt/tag/idaeh/), and the Pacunam: Fundación Patrimonio Cultural y Natural Maya (7a. av. 6-53, Zona 4. Ed. El Triángulo, 5o. Nivel, Of. 5B, Ciudad de Guatemala, Guatemala C.A., https://pacunam.org/), respectively. Researchers interested in these lidar data sets should contact the institutions listed above.

**Funding:** This study was funded by the Research Center of the Slovenian Academy of Sciences and Arts (ZRC SAZU; https://www.zrc-sazu.si/en/node), the Slovenian Research Agency (https://www.arrs.si/en/; research program ARRS P6-0079), the Instituto Nacional de Antropología e Historia (INAH), Mexico (https://www.inah.gob.mx/), and FBC Datec company, Mexico.

**Competing interests:** The author has declared that no competing interests exist.

complexes of this type, commonly labeled as Group E-type complexes, E-Group assemblages, or simply E Groups, had a particularly prominent role in Maya astronomical observations. Some scholars have argued that these complexes were merely commemorative or allegorical allusions to celestial cycles, without any observational function, but all these hypotheses reflect the belief that the cosmological and astronomically-derived concepts were associated specifically with E Groups [3]. In order to test the highly divergent hypotheses, most of which are based on unreliable alignment data, I recently undertook measurements in a number of E Groups in the central Maya Lowlands, characterized by the largest concentration of these compounds (Fig 1).

A typical E Group has a symmetrical ground plan, with its central axis running approximately east-west, from a pyramidal temple on the west side of a plaza to the center of an elongated platform that delimits the plaza on its east side and extends in a roughly north-south direction. The latter has no superstructures in the so-called La Venta type, which is the earliest variant of this assemblage. The Cenote type has a central building on the platform, slightly set backwards, whereas the Uaxactún type, the latest version of the complex, has two additional buildings on the platform's extremes; all E Groups measured for the purposes of this study are of the latter two types. In every E Group I measured three sightlines typically interpreted as astronomically significant; one is the central axis, running from the western pyramid to the central structure on the eastern platform, and the two others, hereinafter briefly referred to as lateral alignments, connect the western pyramid with the northern and southern extremes of the eastern platform (Fig 2).

E-Group assemblages are dated largely to the Middle and Late Preclassic and Early Classic periods (~1000 BCE– 600 CE) and normally have several construction stages. Most of them are concentrated in the central lowlands of the Yucatan peninsula, but the earliest forms seem to have appeared along the Pacific coast and in the highlands of Guatemala and the Mexican state of Chiapas. In the Maya area, E Groups were one of the earliest standardized forms of monumental architecture. Their role was not everywhere and always the same, as evidenced by regional and time-dependent variations in sizes and shapes of the buildings, as well as by the characteristics of the associated archaeological contexts. Most importantly, they provided a stage for ritual performances, which in communities with an increasingly complex social organization became a significant part of political ideology [4–11]. Given the evident socio-political significance of E Groups, which characterize practically all civic and ceremonial cores of Maya cities in the central Yucatan peninsula, a proper assessment of their possible astronomical connotations is of foremost importance for understanding some fundamental principles of Maya architectural and urban planning and the underlying cosmological systems.

Several decades of archaeoastronomical research in Mesoamerica have shown that the clearly non-uniform distribution of orientations in civic and ceremonial architecture can be explained in astronomical terms. No other possible orientation motive (climate, local topography, magnetism, defensive concerns, etc.) can account for the widespread and long-lasting alignment groups. The only conceivable rationale for the concentrations within certain azimuthal ranges is the use of rising and setting points of celestial bodies as reference objects. It was also argued that the orientations largely refer to sunrises and sunsets on agriculturally important dates [12–17]. More recently, systematic studies in different Mesoamerican regions revealed that, notwithstanding some regional and time-dependent variations, the practice of orienting important buildings was based on the same principles throughout Mesoamerica. The dates most frequently recorded by solar orientations, which prevail, cluster in agriculturally significant seasons, and the intervals separating them tend to be multiples of 13 and 20 days. Since the latter were elementary periods of the Mesoamerican calendrical system, such orientations enabled the use of observational calendars that facilitated a proper scheduling of

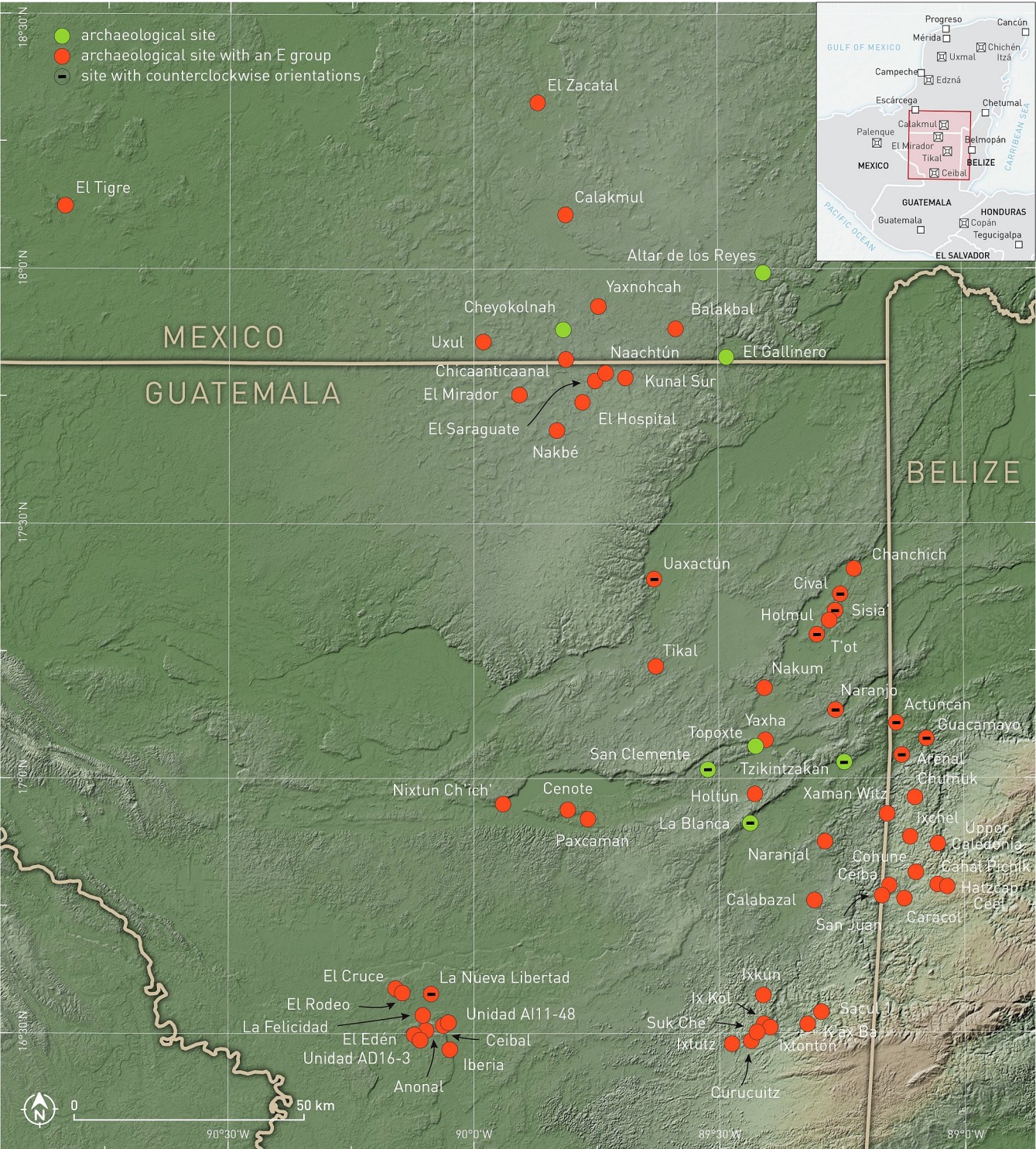

**Fig 1. Map of the central Maya lowlands, with the location of sites included in the study.**

agricultural activities and the corresponding rituals: knowing the mechanics of the formal calendar and the structure of the observational scheme, it was relatively easy to predict the relevant dates (the dates separated by multiples of 13/20 days had the same number/sign of the

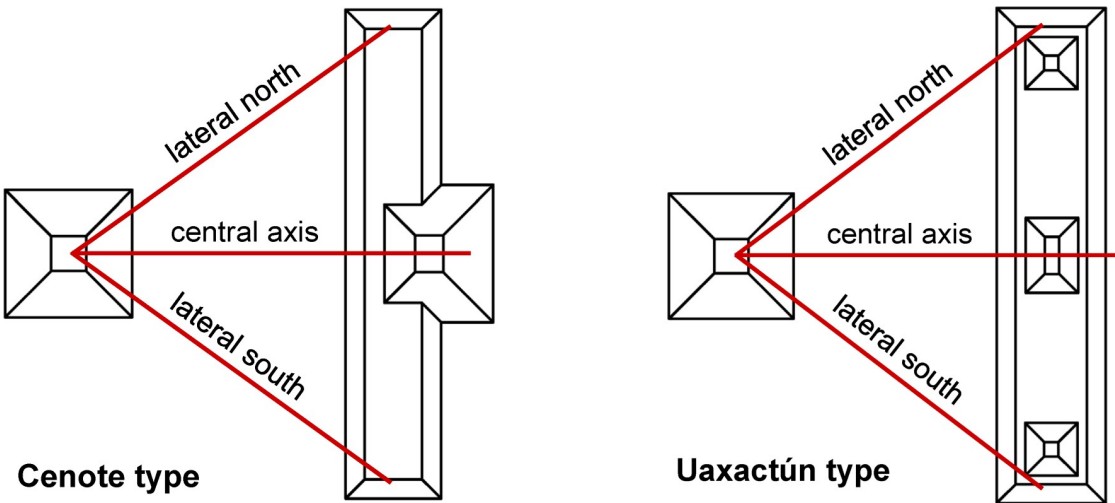

**Fig 2. Schematic ground plans of two most common types of E Groups, showing the alignments that have been measured and analyzed.**

260-day calendrical cycle), even if direct observations were impeded by cloudy weather. As indicated also by ethnographic evidence, this anticipatory aspect must have been an important characteristic of the observational calendars. Considering that modern farmers determine canonical, agriculturally significant dates with the aid of the Western calendar (in some places astronomical observations are still practiced), rather than by observing seasonal environmental changes, it is obvious that the latter are not a reliable reference. Therefore, and given the overwhelming evidence that the Mesoamericans had no regular intercalation system that would have maintained a permanent correlation between the 365-day calendrical and the slightly longer tropical year, astronomical observations were a necessity in prehispanic times [12, 14, 17–24]. The astronomically motivated intentionality of the most prominent alignment groups in the Maya Lowlands has been additionally supported by statistical analyses [25].

## Materials and methods

For any alignment it is relatively easy to find an astronomical correlate, but to propose, with a reasonable degree of confidence, that the observed correspondence is not fortuitous, we need either a statistically significant number of alignments incorporated in a coherent set of archaeological features (i.e. of the same type and pertaining to the same cultural complex) and referring to the same position (declination) on the celestial sphere, or independent contextual evidence suggesting an astronomical motive for the alignment in question (iconography, written sources etc.), or both [13, 26–28]. Regarding E Groups, as well as Mesoamerican buildings in general, contextual data supporting an astronomical rationale for a particular alignment are often ambiguous and, in most cases, nonexistent. Accordingly, with the objective of collecting a sufficiently large sample of reliable quantitative data, I measured alignments in 71 E Groups in the central lowlands and—for comparative purposes—orientations of 79 structures of other types in the same area (Fig 1, S1 Table).

In order to identify possible astronomical targets of alignments, the corresponding declinations were calculated. The declination, a celestial coordinate expressing angular distance from the celestial equator to the north and south, depends on the azimuth of the alignment (angle in the horizontal plane, measured clockwise from the north), geographic latitude of the observer, and the horizon altitude corrected for atmospheric refraction. The alignment data were

acquired in field or, in some cases, with the aid of lidar imagery. In field measurements and data reduction, the methods and procedures established in archaeoastronomical work were employed [12, 20, 23, 26, 27]. For the alignments within the arc of solar movement along the horizon, the corresponding sunrise and sunset dates and the intervening intervals were determined. All these data were assigned errors derived from the estimated uncertainties of azimuths obtained with measurements. In order to assess the intentionality of alignments and their possible astronomical referents, the method known as kernel density estimation (KDE) was employed in the analyses. The advantage of this method over the use of simple histograms is in that the errors assigned to similar values tend to cancel out; it can thus be expected that the most prominent peaks of the resulting curves, which present relative frequency distributions, closely correspond to the values targeted by particular orientation groups. Since the most conspicuous peaks, as will be seen, manifest a good agreement with those obtained in former studies in the Maya Lowlands, they allow similar interpretations of their significance, but also disclose variations specific to the analyzed data set; on the other hand, the lack of significant clustering of values corresponding to certain types of alignments makes incredible some previous hypotheses. For details on methods and techniques, see S1 Text, in which tabulated alignment data for all sites included in the study are also given (S1 Text; S1 Table).

## Analyses of alignment data

### Azimuths

The azimuths of the east-west axes of symmetry of E Groups and other types of buildings in the central lowlands exhibit similar frequency distributions, while the distribution of lateral alignments of E Groups is different and more dispersed (Fig 3). The central axes of E Groups and buildings of other types also share a predominant clockwise skew from cardinal directions (south-of-east/north-of-west). This trend, observed throughout Mesoamerica [12, 14, 16, 20], is attributable to the symbolic connotations of the east and west. As a consequence of the south-of-east deviations, the dates recorded by solar orientations on the eastern and western horizons fall mostly in the dry and rainy season, respectively, which is consistent with the evidence suggesting that the dry season was conceptually related to the eastern part and the rainy

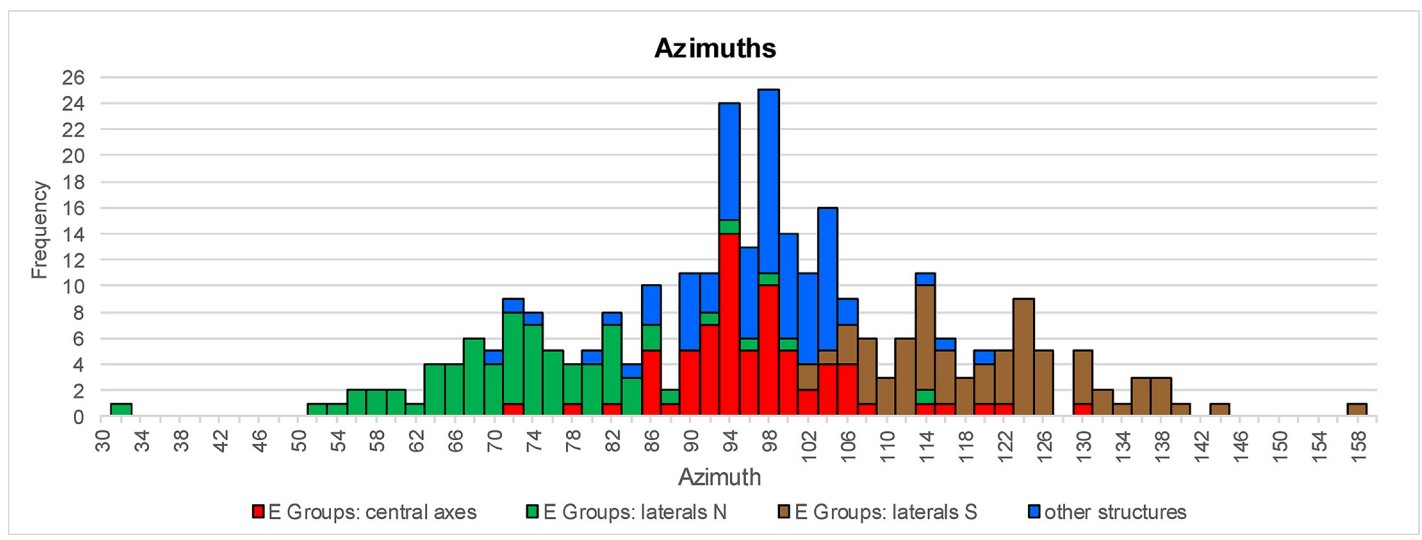

**Fig 3. Frequency distributions of azimuths of alignments in E Groups and of east-west azimuths of buildings of other types included in the study.**

season to the western part of the universe. Particularly revealing are the symbolism and directional associations of the Sun, the Moon, and Venus; the Sun, presiding over the east, was related to heat, fire, and drought, while the Moon and Venus, particularly its evening manifestation, were linked with the west and with water, maize, and fertility [21, 29].

However, a zone in eastern Petén and western Belize is characterized by orientations skewed counterclockwise from cardinal directions. Aside from the sites shown in Fig 1, others in the same area exhibit the same peculiarity (e.g. Xunantunich, Baking Pot, Buenavista del Cayo, El Pilar, Pacbitun), obviously reflecting a regional tradition, which can be explained in terms of political relations starting in the Preclassic and continuing in later periods, as suggested by other types of archaeological evidence, including hieroglyphic texts [30, 31].

## Declinations

Putative astronomical target(s) of an alignment can be identified only by determining the corresponding declination. Fig 4 shows relative frequency distributions (KDE) of declinations

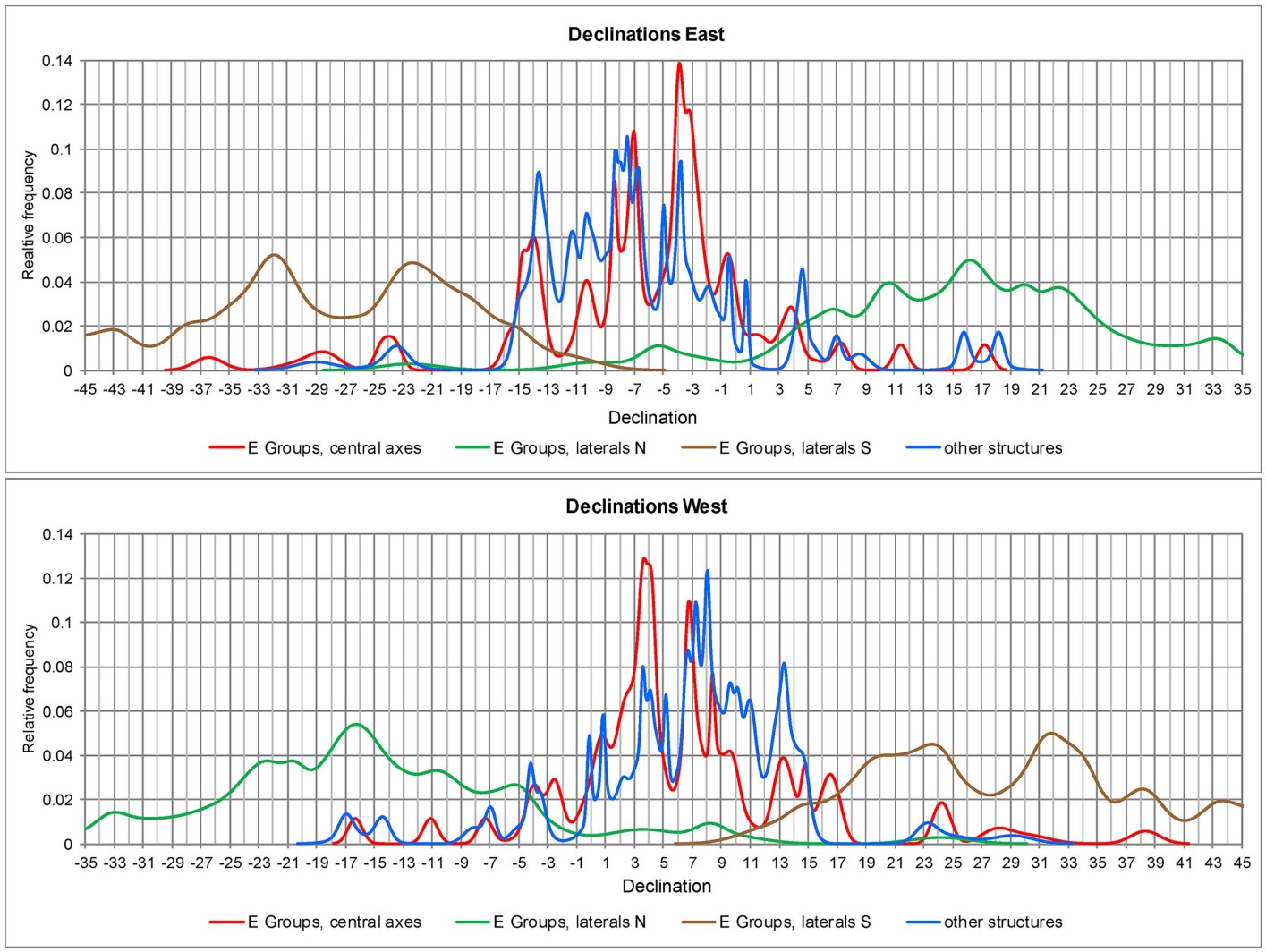

**Fig 4. Relative frequency distributions (KDE) of declinations.**

marked on both horizons by alignments in E Groups and by orientations of other types of buildings in the central lowlands. Regarding astronomical use of E Groups, it is generally assumed that, observing from the western pyramid, the central mound on the eastern platform and its northern and southern extremes marked astronomically significant positions on the eastern sky. However, western directionality of these alignments cannot be discarded, considering that the orientations of the Maya and other Mesoamerican buildings were astronomically functional either to the east or to the west, or sometimes in both directions. It has also been shown that the placement of the access or main façade does not necessarily indicate the astronomically functional direction, but was likely conditioned by the beliefs and rituals associated with a particular structure and the symbolism of world directions [18, 20, 21]. The consistent arrangement of E Groups, with a pyramid on the west and a platform on the east side, can be explained in the same terms, rather than as indicative of directionality of alignments.

The heights of buildings integrated in E Groups block the view to the natural horizon along various alignments, particularly along lateral lines in western direction, because the western pyramid is generally higher than the eastern platform. However, all declinations have been calculated for the natural horizon altitudes, both because in most cases the original heights of buildings and, therefore, the corresponding declinations cannot be reliably established and because the natural horizon would have likely been visible in early stages of every E Group. At Caracol, for example, the early stages of the E Group are from the Late Preclassic, but the western pyramid (Structure A2) did not reach its current height until the Late Classic [5, 32]. In fact, it is unlikely that astronomical phenomena were observed on the artificial horizon shaped by buildings, because such alignments would have been of low precision: due to the relatively short distances, the corresponding sunrise or sunset date, or any other celestial phenomenon, would have depended on the exact observer's position and even body height. Furthermore, as the distribution patterns demonstrate, the architectural alignments in Mesoamerica, as a rule, referred to astronomical phenomena on the natural horizon [12, 14, 17, 21].

As evident in Fig 4, the declinations marked by the central axes of E Groups and other building types exhibit similar distributions, and some concentrations even have almost identical peaks, suggesting an astronomical rationale. In contrast, the declinations marked by lateral alignments are spread over a much wider angle and their distribution does not exhibit a pattern for which a convincing astronomical explanation could be proposed. Two peaks, however, are relatively prominent and deserve some attention.

In the curve showing the distribution of declinations marked by south laterals on the eastern horizon, the peak at the value of about -22.5˚ might refer to the December solstice sunrise. The other peak at the value of ca. -32˚ might only refer to a star or asterism. Among the relatively bright stars, Shaula ($\lambda$ Scorpii) or Kaus Australis ($\varepsilon$ Sagittarii) are possible candidates. Given their magnitudes (1.6 and 1.8, respectively), their extinction angle (the minimum vertical angle above the mathematical horizon at which a star is visible) in a humid atmosphere near the sea level is about 4˚ [33]. Replacing horizon altitudes with this angle, the declinations of the alignments in question would change and the peak in the curve would move to about -31˚. Both Shaula and Kaus Australis had approximately this declination around 400 BCE [34]. A number of buildings on the northeast coast of the Yucatan peninsula may have targeted an asterism in that part of the sky, as suggested by both alignment patterns and some contextual evidence [19]. However, since no buildings oriented in that direction are known in the central Maya Lowlands, and in the absence of independent data to the contrary, a stellar motive for the lateral alignments in E Groups remains questionable.

Since there is no evidence suggesting a prevalent importance of the southern extreme of the eastern platform, it is difficult to explain why only that alignment would have been astronomically functional. And if we suppose that both laterals were astronomically motivated (the lack

of their patterned distribution being due to the low precision of alignment data), such a conjecture is difficult to reconcile with the layout of E Groups: as discussed below, their central axes were clearly laid out on astronomical grounds, but most of them are skewed south of east; if also the extremes of the eastern platform had been intended to mark certain astronomical phenomena, they would have hardly been as symmetrical to the central axes as they are in most cases. The declinations corresponding to many lateral alignments are beyond the solar span (ca. ±23.5˚), but—given the prevalent south-of-east skew of the central axes—not in the same proportion (57% of south laterals but only 14% of north laterals lie beyond this angle).

In sum, the possibility that some lateral alignments were astronomically motivated cannot be discarded. However, given their unpatterned distribution, as well as the absence of independent evidence suggesting their importance (see below), their astronomical intentionality is, in general, unlikely. They will thus be excluded from further analyses.

On the contrary, the distribution pattern of declinations corresponding to the central axes can be convincingly explained in astronomical terms. Only three of these alignments are placed beyond the angle of solar movement along the horizon. Plaza D of Sacul 1 has no readily apparent astronomical referent, while E Groups of Chumuk and Xaman Witz may have been oriented to the major southerly/northerly extremes (standstills) of the Moon on the eastern/western horizon (S1 Table). Lunar orientations are not uncommon in the Maya area [35]. They have not been reported in the central lowlands, but the correspondence of Chumuk and Xaman Witz E Groups with lunar extremes may well be intentional, considering that both sites are located in the area north of Caracol, where a woman with accoutrements of the goddess Ixchel, associated with the Moon, was buried in the Northeast Acropolis around AD 150 [32, 36] and the upper surface of Altar 25, probably from the Terminal Classic, features a lunar glyph with a female figure interpreted as a lunar goddess [37]. Besides, another orientation fitting major lunar extremes is materialized in a triadic complex at Ceiba, connected to the rest of Caracol with a causeway [38, 39]. Significantly, the same locale has an E-Group with a solstitially aligned central axis (S1 Table); associations of solstitial and lunar orientations are common along the northeast coast of the Yucatan peninsula, suggesting that full Moon extremes were observed [19, 35].

## Dates and intervals

The central axes of most E Groups can be related to the Sun's positions on the horizon. Due to the errors estimated for individual alignments, the intended dates cannot be determined in every particular case. However, and although not necessarily each and every E Group was oriented on astronomical grounds, the peaks in Figs 5 and 6 clearly reflect the purpose of recording certain dates and intervals on both the eastern and the western horizon (designated briefly as east and west dates and intervals). Western directionality is additionally supported in several E Groups where the central building on the eastern platform is higher than the western pyramid [6]. Moreover, we cannot be certain that it was precisely the line connecting the western pyramid and the central eastern building that was observationally functional; the relevant alignment could have been determined by the very orientation of either of the two structures (this is particularly likely in the not so few cases where the western pyramid is notably higher than the opposite structure, which would have therefore been a very poor marker of the Sun's positions on the natural horizon lying well above it). In most cases it was only possible to measure the axis connecting the summits of the western pyramid and the central structure on the eastern platform, because the present state of the buildings integrating E Groups makes it impossible to determine their orientations. Nonetheless, wherever these buildings are sufficiently well preserved, their orientations largely agree with the central axes (in other words, they are oriented to each other, emphasizing the importance of the alignment they compose)

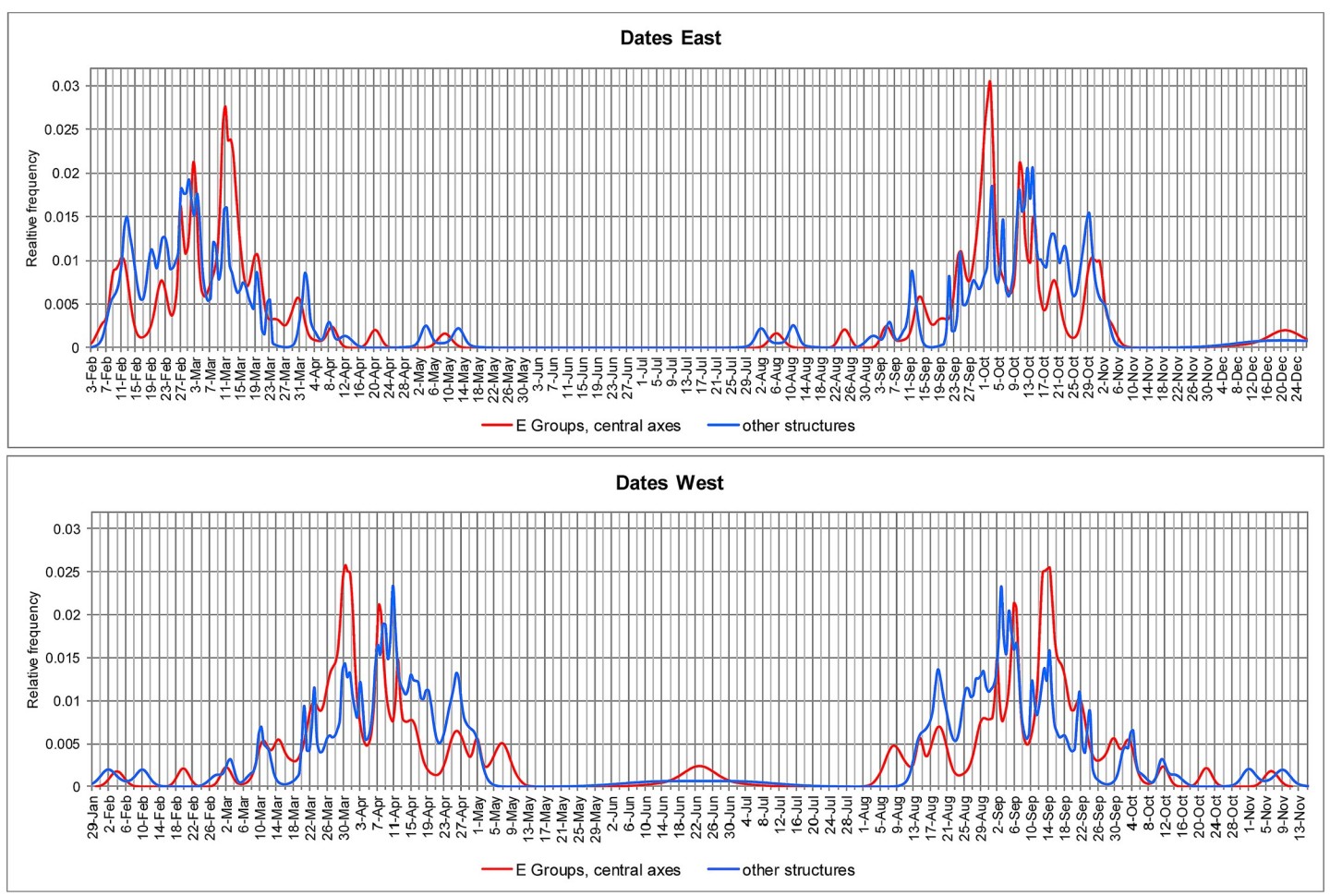

**Fig 5. Relative frequency distributions of dates marked by solar alignments.**

and the analyses of the data indicate that the latter were, in general, observationally functional, which means that, for an observer standing on one structure, the opposite one served as a fore-sight, facilitating observations.

Figs 5 and 6 show that the distributions of dates and intervals corresponding to E Groups and other structures are similar; the more pronounced peaks in the case of other building types are due to the fact that their orientations, as a rule, can be measured with better precision, resulting in smaller estimated errors. In agreement with the results of former studies in the Maya area and elsewhere in Mesoamerica, Fig 6 demonstrates the importance of dates separated by calendrically significant intervals (multiples of 13 and 20 days). It should be recalled that any solar (except a solstitial) alignment matches two sunrise and two sunset dates and each pair of dates divides the year into two complementary intervals whose sum is equal to the length of the tropical year.

The most prominent peak in the curve showing the distribution of east dates targeted by E Groups corresponds to March 11 and October 2 (Fig 5; declinations near -4°: Fig 4), separated by an interval of 160 (= 8 × 20) days; accordingly, this interval is marked almost exactly by the highest peak in the curve of east intervals shown in Fig 6. Many structures of other types are also oriented to sunrises on these dates, both in our study area (Figs 5 and 6) and elsewhere in the Maya Lowlands [18].

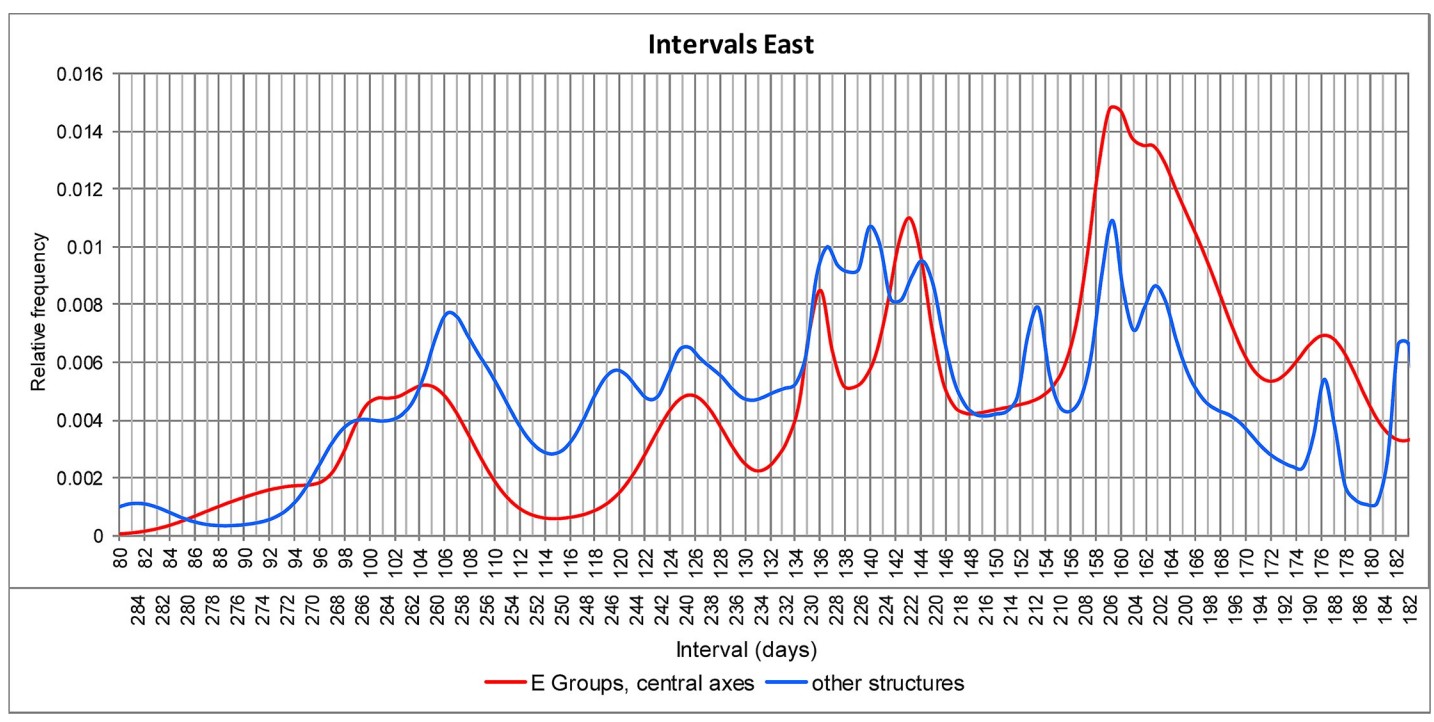

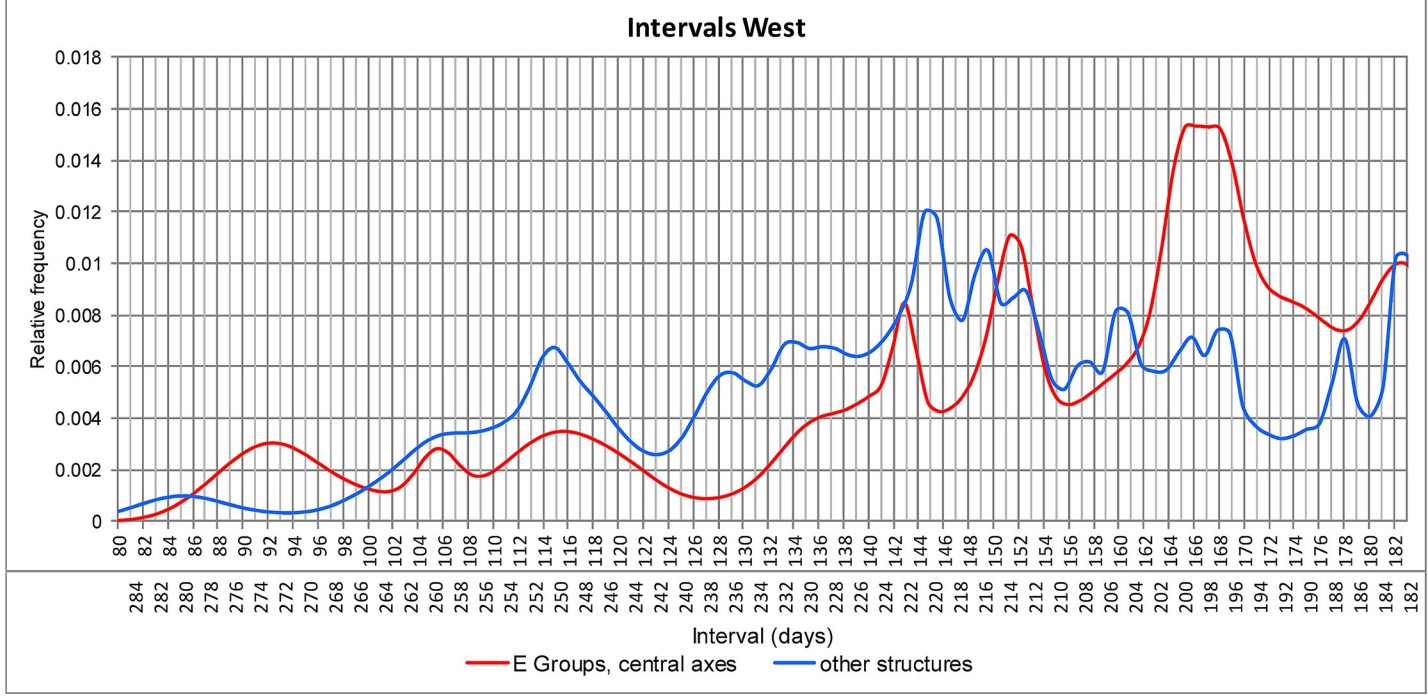

**Fig 6. Relative frequency distributions of intervals separating the dates recorded by solar alignments.**

The orientations marking sunrises on March 11 and October 2 correspond to sunsets around March 31 and September 12, separated by an interval of 200 (= 10 × 20) days (declinations around 4˚). Some of these alignments could have been functional to the east and others to the west. Moreover, a few alignments—due to appropriate horizon altitudes—could have

been functional in both directions, e.g. at Nixtun-Ch'ich' and Hatzcap Ceel. It is precisely this fact that may account for the widespread importance of this orientation group.

Some alignments that are skewed counterclockwise from cardinal directions (north of east/ south of west) and are common in the border zone between eastern Petén and western Belize (Fig 1) mark the same pairs of dates, but on the opposite horizons. For example, the E Group of Guacamayo matches sunsets on March 11 and October 2, whereas the E Groups of T'ot and Sisia' may have recorded sunrises on March 31 and September 12, or perhaps sunsets on March 11 and October 2. The orientation of Structure A-18 of Uaxactún could have been functional in both directions, recording sunrises and sunsets on the same pairs of dates.

The importance of the 160-day and 200-day intervals is additionally attested in alignments that mark other dates. The North and Far West E Groups of Cival are oriented to sunsets on March 14 and September 30, separated by 200 days, whereas various buildings recorded the same dates on the eastern horizon (e.g., North Plaza E Group and Structure I of Naachtún, Maler Group of Yaxhá, E Group of Curucuitz). The E Group at Guacamayo was likely functional in both directions, recording sunsets on March 11 and October 2 and sunrises on April 3 and September 10; both pairs of dates are separated by an interval of 160 days. Sunrises on April 3 and September 10 also correspond to the orientations of Structures XIV and XV (twin pyramids) of San Clemente (and probably of Structures III and VII of the Palace of this site), whereas Building Y of Nakum, the tallest of the site, is oriented to sunsets on the same dates.

While one group of orientations marked the 200-day interval on the western horizon, a nearby peak in Fig 6 (lower graph) suggests that in some cases the purpose was to achieve the interval of 169 (= 13 × 13) days, delimited by sunsets on March 29 and September 14. However, due to possible errors of the alignment data, the intended target cannot be ascertained in every particular case.

Another group of orientations materialized in E Groups and other buildings corresponds to sunrises on March 2 and October 10 (declinations near -7°) and sunsets around April 7 and September 6 (declinations around 7°). Since the intervals separating the latter pair of dates (151 and 214 days) do not seem significant, these orientations must have been astronomically functional to the east, marking sunrises on March 2 and October 10, separated by 143 (= 11 × 13) days and recorded by various orientations in the Maya area [18, 40]. The importance of this interval is also indicated by the peak in the curve showing the distribution of west intervals (the east intervals corresponding to the same orientation group concentrate around 136/ 229 days), delimited by sunsets on April 11 and September 1, though the sunsets on March 2 and October 10, probably recorded by the E Group of La Nueva Libertad (periphery of Ceibal), skewed north of west, also contribute to this peak.

One group of alignments incorporated in E Groups and other buildings corresponds to sunrises around March 20 and September 25 and to sunsets around March 23 and September 21, with the intervening intervals of 176/189 and 182/183 days, respectively. While the first pair of dates has no conceivable significance, March 23 and September 21, ±1 day (declinations near 1°) are the dates that divide each half of the year delimited by the solstices into two equal periods and are commonly labeled quarter-days. A number of orientations in the Maya area and elsewhere in Mesoamerica match sunsets on these dates [21]; however, the E Group and Structure B-15 of Naranjo and, possibly, the main E Group of Cival, which are skewed counterclockwise from cardinal directions, marked these dates on the eastern horizon.

A few other peaks in Fig 6 are less prominent, but also suggest the importance of calendrically significant intervals. The one of 260 days is delimited by sunrises on February 12 and October 30, while the interval of 240 days separates sunrises on February 22 and October 20. These dates are marked by the two most widespread orientation groups in the Maya Lowlands,

particularly prominent being the first one, most likely because the phenomena separated by 260 days occurred on the same date of the ritual calendrical cycle [18, 21].

## Contextual support

In the light of the analyses presented above, the central axes of most E Groups were laid out on astronomical grounds, whereas the lateral lines, notwithstanding some possible exceptions, did not have any astronomical significance. Various types of independent evidence, discussed below, are consistent with this conclusion, indicating the importance of the central axes and the lack of attention paid to the lateral alignments.

At several sites, various buildings or architectural complexes not only reproduce the orientations of E groups but are also placed along their central axes, sharing their orientation. The most prominent cases are found at Nixtun-Ch'ich', Yaxhá, Cival, Caracol, and Naachtún (see details below). In contrast, no structures located along the lateral lines of these complexes have been identified.

During excavations in several E Groups, offerings and burials have been found along their central axes, for example, at Tikal [41, 42], Caracol [5], Ceibal [43, 44], Nakum [45], and Cival [46], but not along the lateral alignments. Among the offerings located along the central axis of the Lost World E Group of Tikal, there were even ceramic pieces with designs of possible astronomical connotation [42]. It should be clarified that such offerings were clearly aligned, often extending to the plazas, whereas the deposits in the extreme buildings of eastern platforms, also found in several E Groups, were placed on particular spots, without being spatially arranged along lateral alignments.

Where excavations have established it, the central axis of the E Group was preserved during various construction stages, but lateral alignments were not, because the eastern platform changed its dimensions or even its position. In Tikal, the east-west normative axis of Lost World did not change during all its construction stages from the Middle Preclassic on, but the length of the eastern platform increased over time [42]. At Ceibal, the currently visible E Group in the Central Plaza, with Structures A-9, A-10 and A-12 on the eastern platform and the pyramidal Structure A-20, is from the Classic period, but its earliest stage dates to about 950 BCE (Real 1 phase of the Middle Preclassic). The central axis of the complex was preserved throughout its long history, whereas both the position and the dimensions of the eastern platform changed [43, 47]. Likewise, the central axis of the E Group at Caracol was maintained from the Late Preclassic to the Late Classic, but the consecutive construction stages of the eastern platform had different sizes [5].

## Discussion of previous hypotheses

The opinion that Group E of Uaxactún, the first archaeologically explored architectural complex of this type, was a solstitial and equinoctial observatory [1, 2] became widely accepted and the same use was attributed to other similar assemblages discovered later at other sites. Though it was soon evident that most of their orientations are notably different [48, 49], it is remarkable how often we can still read, even in most recent scholarly literature, that they served for observing the equinoctial and solstitial sunrises. In general, the solstices and equinoxes are often mentioned in tandem, as if they were the only conceivably significant moments of the tropical year. In fact, while the solstices are marked by easily perceivable extremes of the Sun's annual path along the horizon, the equinoxes are not directly observable. The equinox has a precise meaning within the framework of Greek geometrical astronomy that underlies the Western scientific tradition, which defines the equinox as the moment when the Sun crosses the celestial equator, having the declination of 0˚. But since the celestial equator is a theoretical

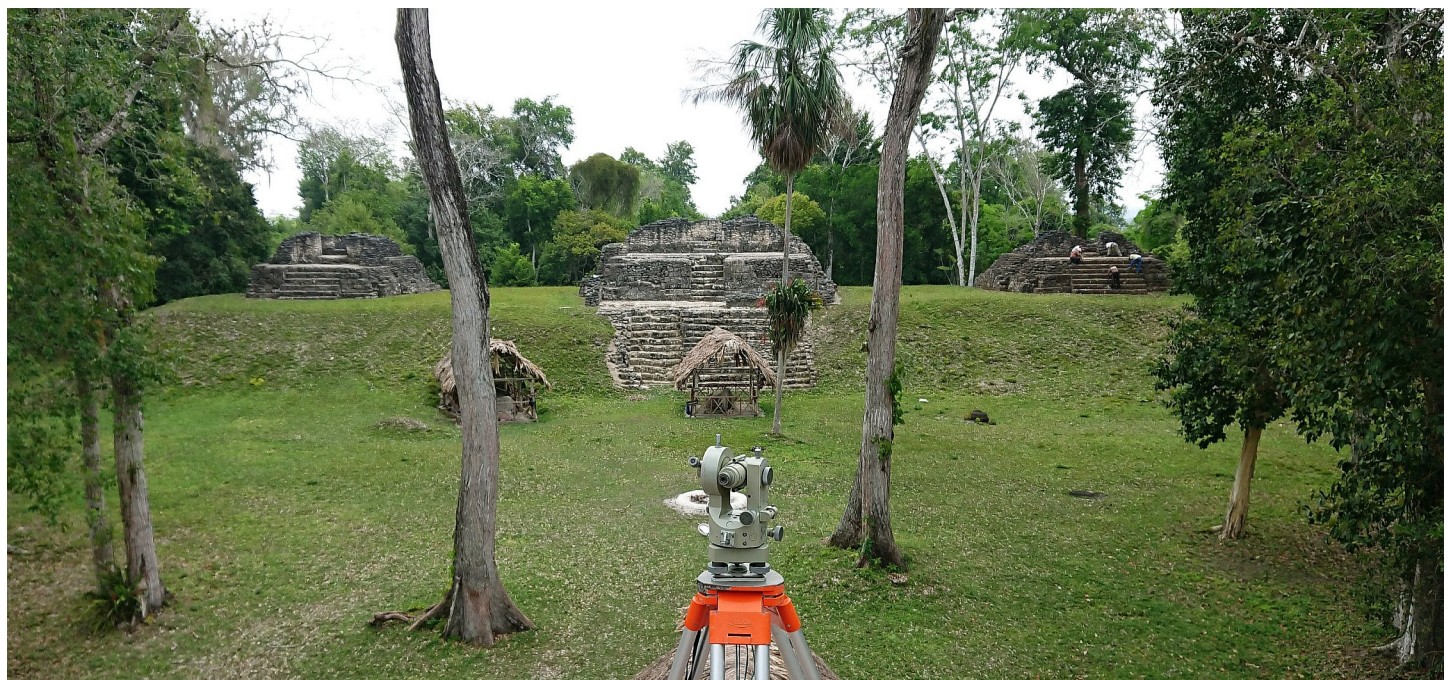

**Fig 7. Group E of Uaxactún, east platform with Temples E-1, E-2 and E-3, viewing east from the western pyramid (Structure E-7-sub-2).**

geometric construct based on a specific celestial coordinate system, it is utterly unlikely that identical concepts would have developed independently in other ancient societies [50–52]. Epigraphic records and ethnographically documented survivals of prehispanic concepts offer no compelling evidence that the Maya and other Mesoamericans were aware of the equinoxes, and the declinations corresponding to architectural orientations manifest no clustering centered on 0˚; instead, the most likely target of the near-equinoctial alignments were the so-called quarter-days, which fall two days after/before the spring/fall equinoxes [21, 22], and the same conclusion, as argued above, applies to E Groups (Figs 4 and 5).

Contrary to the very popular idea, not even Group E of Uaxactún (Fig 7) can be related to the equinoxes. Assuming that the lateral alignments marked the solstices, Ricketson [1, 2] established that the most convenient observation point would have been on top of the earliest stage of the western pyramid (Structure E-7), but he also noted that the equinoctial line does not pass through the center of the central building on the eastern platform (Structure E-2) but rather over the northern jamb of its entrance. Aveni and Hartung [48] confirmed the location of the most appropriate observation point and also noted that Group E could have served as an approximate solstice observatory, but not as an equinoctial one, because the Sun on the equinoxes would have appeared above Structure E-2 and to the right of its center. Later excavations revealed that the alignments proposed to have been astronomically significant connect buildings from different periods and thus could not have been observationally functional [53]. The earliest version of the western pyramid (Structure E-7-sub-1), considered to have been the most convenient observation point, was built at the beginning of the Late Preclassic (300–100 BCE), but was covered by the one currently exposed (Structure E-7-sub-2) already by the end of the Late Preclassic (100 BCE– 100 CE). It was not until the Early Classic (300–378 CE) that the eastern platform reached its current dimensions and the lateral temples were built, but during that period the western pyramid (Structure E-7, destroyed during excavations in the 1920s) reached a height of about 15 m [1, 5, 54–56]. In other words, when the eastern platform

with upper buildings were given the shape and layout visible today, the "ideal" observation point had long been covered; an observer on top of E-7 would have seen solstitial sunrises considerably north and south of Temples E-1 and E-3, and the equinox Sun would have risen north of the center of Temple E-2 (note that, therefore, the alignment data in S1 Table for Group E of Uaxactún are irrelevant, because they are valid for an observer on top of the currently exposed Structure E-7-sub-2).

Even for the central axis, such as can be measured today, there is no readily apparent astronomical rationale; neither the corresponding sunrise nor sunset dates seem significant. Therefore, either the whole compound in its latest version had no astronomical function, or the top of (the now disappeared) Structure E-7 was not placed along the central axis measurable today. It is possible that the central axis of earlier versions of the complex was slightly different from the current one, but cannot be reconstructed because the excavation reports do not provide the exact location of the central structure on the earlier versions of the eastern platform. If it was slightly to the north of the late Structure E-2, the central axis of the group may have recorded quarter-day sunrises. Such a scenario is speculative, but is at least suggested by the currently exposed Structure E-7-sub-2, which is skewed about 1° counter-clockwise from cardinal directions, as well as by the similar orientation of Group D, another Group E-type complex at Uaxactún.

Soon after the first descriptions of Group E of Uaxactún, Ruppert [49] noticed that other compounds of this type have different orientations and suggested that they had more ceremonial than observational functions. Interpreting them as allegorical imitations of the astronomically functional template at Uaxactún, Vilma Fialko [57] labeled them Complexes of Astronomical Commemoration. Although it soon became evident that many E Groups were earlier than the supposed prototype at Uaxactún, it was still argued that their initial purpose was to record the solstices and equinoxes, but that this observational function was subsequently abandoned and replaced by a predominantly ritual use [58, 59]. It was also suggested that stone columns or wooden poles placed on the eastern platform could have served as markers of the equinoxes or solstices [58, 60], but no specific evidence supports this hypothesis. All these conjectures share the prejudice that the only potentially significant moments of the tropical year were the equinoxes and the solstices.

Aveni and Hartung [48] also proposed that most E Groups were nonfunctional imitations of Group E of Uaxactún, but this hypothesis was abandoned by Aveni et al. [61], who argued that both central axes and lateral alignments enabled the use of observational schemes composed of calendrically significant intervals. Analyzing sunrise dates, they noted that three frequently recorded date pairs (February 19/October 22, March 11/October 2, March 31/September 12) are separated by multiples of 20 days (80, 60 and 40 days, respectively) from the days of the zenith passage of the Sun in latitude 17.5°N (May 10/August 3), the mean latitude of the sites included in their study. Although in several cases their alignment data (often based on inaccurate site maps) do not agree with ours, our data sample corroborates the frequency of these dates (Fig 5). However, their significance can hardly be related to the zenith passage dates in central lowland latitudes, because the same dates are also recorded by a number of orientations elsewhere in the Maya Lowlands and in other Mesoamerican regions [18, 20, 21, 24]. Rather, the importance of these date pairs can be explained by the fact that they delimit multiples of 20 days (120 days from October 22 to February 19; 160 days from October 2 to March 11, 200 days from September 12 to March 31). As shown in Fig 8, the intervals connecting the sunrise dates recorded by alignments in E Groups with the nearest zenith passage dates, which were determined for the specific latitudes of the corresponding sites, exhibit a dispersed distribution, without any notable concentrations around calendrically significant multiples.

Aveni et al. [61] also suggested that the practice of targeting dates related to zenith passages replaced an earlier observation scheme based on the solstices, because the alignments to the

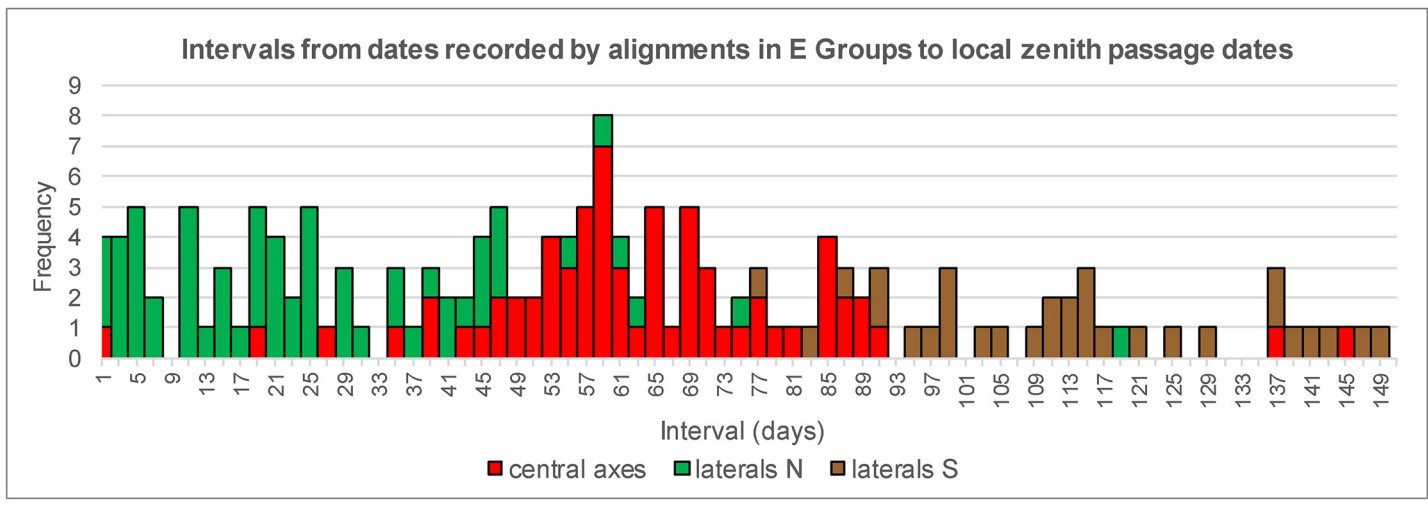

**Fig 8. Frequency distribution of intervals from sunrise dates marked by alignments in E Groups to nearest zenith passage dates.**

solstitial positions of the Sun tend to occur in earlier E Groups, and considering the chronological priority of solstitial orientations elsewhere in Mesoamerica. The hypothesis is based on a few, mostly lateral alignments in E Groups. However, if we only consider the central axes, which were clearly based on astronomical criteria, only two of them in our data sample can be related to the solstices (at Ceiba and Ixchel), whereas various E Groups with construction stages reliably dated to the Middle Preclassic have different orientations, including the one at Ceibal, the earliest example known so far. Accordingly, while solstitial orientations are relatively common in the Maya Lowlands, characterizing different types of structures, they are rarely incorporated in Preclassic architecture [18, 21]. Therefore, currently available evidence does not corroborate a chronological priority of solstitial alignments in the Maya Lowlands.

Ever since Marquina and Ruiz [62] proposed that a number of Mesoamerican buildings were oriented to the Sun's horizon positions on the days of its passage through the zenith, the idea has been very popular. Also relatively widespread is the opinion that the dates of nadir (or antizenith) transit of the Sun were frequently targeted. Both hypotheses have also been applied to E Groups [9, 58]. The Sun's transit through the nadir is not observable; the day of this event could only have been determined indirectly, employing different procedures with varying degrees of precision, but there is no convincing evidence that the Mesoamericans attempted to achieve this goal [18]. On the other hand, various kinds of data indicate the importance of solar zenith transits, which were probably observed by means of devices that allowed the passage of solar rays at noon. However, an analysis based on a large number of orientations revealed no significant correspondence with the Sun's positions *on the horizon* on the nadir and zenith passage dates [63], and the same conclusion applies to E Groups. The date of the Sun's passage through the zenith or nadir can be defined as the day when the difference between the absolute values of the Sun's declination and the local latitude comes to be nearest to 0˚. Fig 9 shows correlations between the declinations marked by alignments in E Groups and the corresponding latitudes (given the range of latitudes of E Groups included in the analysis, only a limited extent of declinations is shown). Since the diagonal lines (dashed) connect the points with equivalent absolute values, the alignments corresponding to the few declinations placed on or (considering their possible errors) near this line in the top/bottom graph may have targeted the antizenith/zenith passage dates. However, the lack of any notable concentration and the overall dispersion of declinations make such an intent improbable.

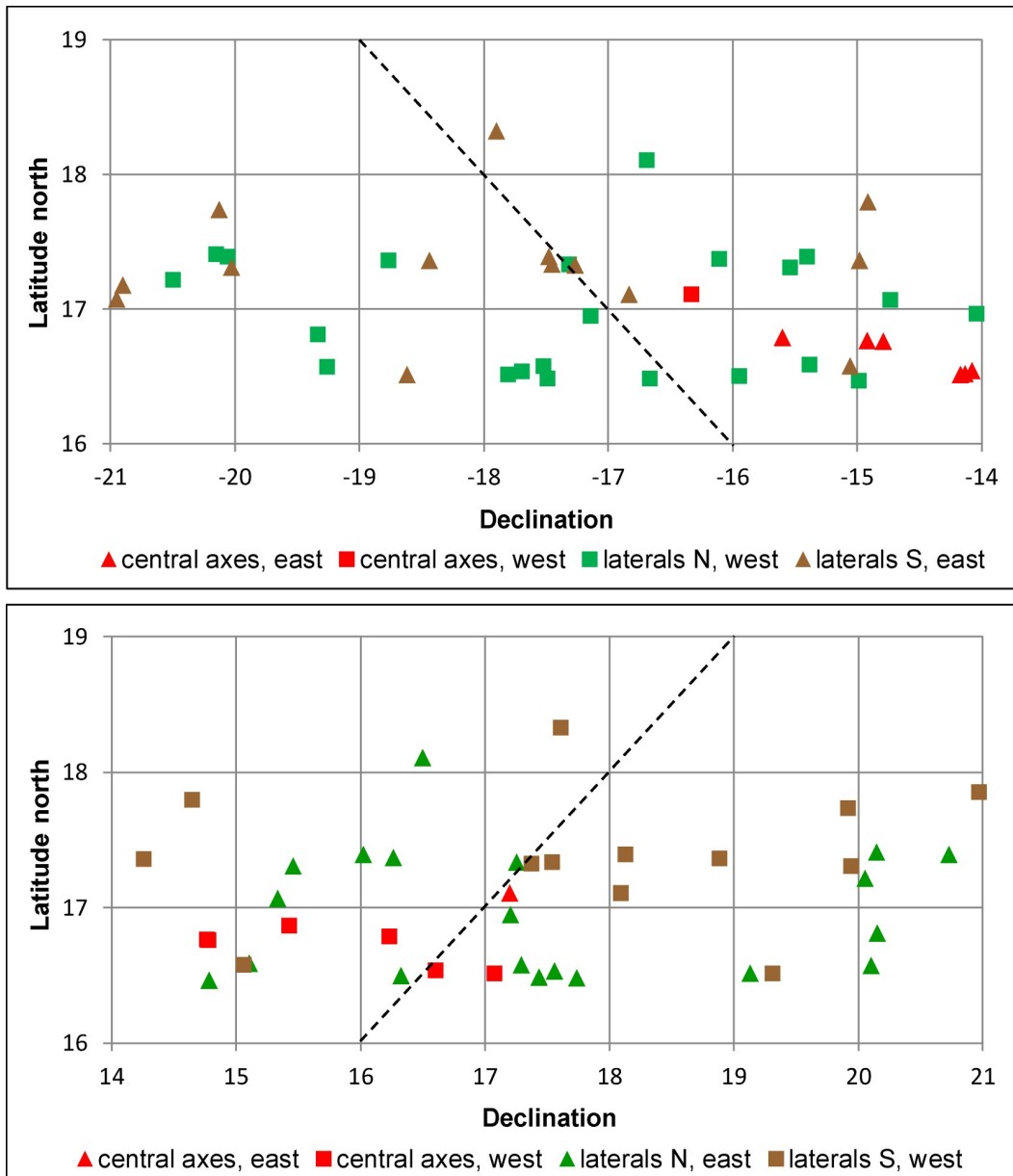

**Fig 9. Correlations between declinations recorded by alignments in E Groups and latitudes of sites.**

It has been noticed that the central axes of some E Groups pointed in the direction where the three stars of the Orion's belt were rising [11, 61, 64]. The intentionality of such correspondences is unlikely, considering that the declination of these stars changed from about -8° to about -4° during the Preclassic period (~1000 BCE– 300 CE), when most E Groups were built, whereas the declinations marked by the central axes, rather than reflecting these precessional shifts, exhibit clear concentrations (Fig 4), for which the Sun is the only conceivable rationale.

While some interpretations rely, to a greater or lesser extent, on those summarized above [65, 66], others are less specific, claiming that E Groups served not only for pinpointing the Sun's positions on the horizon but also for tracking the movement of other celestial bodies within the zodiacal belt [64, 67, 68]. While it is obvious that, observing from the western

pyramid, various celestial objects would have risen above the eastern platform (in many cases not only those within the zodiacal belt), it is unclear what the utility of these sightings would have been. In fact, such speculations could be applied to any elongated building with another in front. The idea that E Groups were more "theaters" or "planetaria" than observationally functional devices [64] is irreconcilable with the evidently non-random distribution of their central axes, which consistently mark the Sun's positions on certain dates. It seems curious that these proposals, although suggesting that E Groups do not contain accurate alignments, nonetheless associate specifically these complexes with celestial and calendrical cycles. Such interpretations, applicable to any building or architectural assemblage, evidently reflect a deeply rooted but unfounded notion that astronomical observations or astronomically inspired rituals were associated exclusively or predominantly with E Groups.

## Significance of alignments

Given the overall similarity of distribution patterns of dates and intervals corresponding to the orientations of E Groups (central axes) and other types of buildings, it is evident that the same principles and purposes were involved in both cases and that E Groups, just like other important structures, allowed the use of observational calendars intended to facilitate an efficient scheduling of agricultural works and related ceremonies. The symbolism of E Groups, associated with maize, water and fertility [8, 10, 58], is consistent with this interpretation. The importance of certain moments of the tropical year attested in orientations survives in agricultural rituals that continue to be performed on the same or nearby dates, and some communities still use the 260-day calendrical cycle, with its constitutive periods of 20 and 13 days and their multiples, particularly for programming agricultural activities. Abundant evidence to this effect has been presented elsewhere [18, 20, 23, 24] and will not be repeated here. For the sake of illustration, I will only comment upon the dates most frequently recorded in the central Maya Lowlands (Fig 5).

The dates in March and April most likely marked appropriate moments for performing ceremonies whose purpose was to assure the timely arrival of rains, indispensable for planting. Among the Itzá of Petén, for example, the favorite day for early planting of various vegetables is April 15, day of San Toribio, while maize and squash are planted a few weeks later [69]. The dates in September and October, on the other hand, must have been related to the rituals intended to guarantee the ripening of maize and abundant harvest; such is the significance of the feasts of San Miguel, on September 29, and San Francisco de Asís, on October 4, popular in various Maya communities [18]. Also noteworthy is that the 16th-century bishop Diego de Landa mentions that "winter begins on St. Francis day and lasts to the end of March", and that "they also sow about St. Francis day a certain kind of maize which is harvested within a short time" [70].

As suggested by current agricultural practices and the role of Christian dates mentioned above, the dates recorded by orientations had a ritual significance (highlighted by the intervening intervals, which were multiples of constitutive periods of the sacred 260-day cycle), while the exact moments of planting and harvesting depended on varieties of maize and other cultigens, as well as on specific environmental circumstances. It is thus reasonable to assume that different versions of observational calendars, based on the same principles but with slightly different structures and canonical dates, were in use simultaneously. These differences, including time-dependent changes, can be explained as a result of politically motivated innovations. An illustrative example is provided by the long construction sequence of the Templo Mayor in the Aztec capital of Tenochtitlan: since its orientation changed with Phase III, commissioned by Itzcóatl, this shift can be understood as a part of the ambitious program of reforms for which this ruler is particularly well known [71].

Considering that the purpose of rituals was to ensure a proper alternation of seasonal climatic changes and, thereby, a successful agricultural cycle, the corresponding dates had to be determined with precision and due anticipation. An important characteristic of observational calendars must have been precisely their anticipatory aspect, not only in Mesoamerica, as already argued in previous studies [21, 61], but also elsewhere. Eloquent information in this regard is found among the historic Pueblos of the US Southwest. Although the precise timing of various agricultural operations depends on farmers' individual decisions based on the weather and the Moon, the moments appropriate for planting different crops and for performing ceremonies must be announced by the Sun watchers ahead of time. Moreover, the Sun Priest's astronomical predictions must be correct within a day, in order to corroborate the effectiveness of rituals [72].

## E Groups in the context of Maya architecture and urbanism: A diachronic perspective

The first stage of the E-Group at Ceibal, dated to about 950 BCE (Real 1 phase) [44, 47], is not only the earliest complex of this type but also the earliest astronomically oriented construction known so far in the Maya Lowlands. It is noteworthy that all of the currently known and reliably established orientations in the Mesoamerican sites that are more or less contemporary with the Real 1 phase at Ceibal refer to the solstices and the (closely related) quarter-days (San Lorenzo, Laguna de los Cerros, altars in the Cuicuilco pyramid, Chiapa de Corzo and various sites along the Pacific coast) [20, 23, 73]. Although these dates can be interpreted as the most elementary references in monitoring the seasons of the year, solstitial and quarter-day alignments are relatively rare in the Preclassic Lowland Maya architecture [18, 21]. Since the concept of E Group likely originated in the Chiapas highlands and along the Pacific coast [10], where solstitial orientations were common and are also materialized in E Groups (e.g. at Chiapa de Corzo [23]), the orientation trend attested in the earliest E Groups in the Maya Lowlands suggests a local innovation. The Real 1-phase Cache 118 placed along the central axis of the Ceibal E Group and composed of 11 greenstone axes (an additional smaller one was found on a higher level and may not have been part of the original deposit) may have alluded to 143 days, or 11 13-day periods, that separated the sunrise dates recorded by the alignment (March 2 and October 10); however, the fact that they were not parallel to the central axis but pointing in a roughly solstitial direction [43] might represent a reminiscence of an earlier orientation practice.

It seems likely that the Ceibal builders chose March 2 and October 10 as target dates because March 2 (presumably the more important date, anticipating the rainy season) was exactly 20 days before the quarter-day of the year, which in that period fell on March 22. While this date pair is also recorded by a number of later orientations, its early occurrence at Ceibal has important implications. We should recall that the alignments marking dates separated by multiples of 13 or 20 days would have only had sense in combination with the formal calendrical system, particularly with the 260-day count. There is some evidence suggesting that this cycle was in use as early as 950 BCE. The olmecoid paintings of the Oxtotitlan cave in Guerrero, Mexico, dated to the 9[th] or 8[th] century BCE, include a motif possibly representing a date of the 260-day count [74]; moreover, Rice [75] has argued that this cycle was likely invented by the Early Preclassic or even before and that some of its day and number signs were derived from the heads and faces of Preclassic clay figurines, which are often found in the contexts of E Groups.

The emergence of astronomical alignments in that period may well reflect an increased reliance on cultivated crops, most notably maize, after 1000 BCE [76, 77]. Although Kennett et al.

[78], based on isotopic evidence from human skeletons in Belize, argue that maize became a staple crop after 2000 BCE, Inomata et al. [79] interpret Real 1 phase at Ceibal as a period of transition to sedentism and agricultural subsistence. Considering the agricultural significance of architectural orientations, the appearance of such orientations in the Maya Lowlands may well have been a response to the growing need of those societies to schedule their labors and accompanying ceremonies in the year of the seasons.

In this context it is worth mentioning the orientation of the urban layout of La Venta. According to the published data, it is deviated 8˚ counterclockwise from cardinal directions and may have referred to sunsets on March 1 and October 12, separated by 140 days [23]. However, since the orientation of the site core, due to its present state, cannot be precisely determined, it is possible that it actually recorded the same dates as the Ceibal E Group, although on the western horizon (as apparently also did the E Group of La Nueva Libertad near Ceibal). While La Venta, therefore, might represent another early case of this orientation group, the fact that the first construction stages of the E Group of Ceibal are earlier constitutes additional evidence of the complexity of interaction, also indicated by other data, between the Maya area and the Gulf Coast in that period [47, 80].

Representing the earliest formalized plan in Maya architecture [6, 8], E Groups were probably also the earliest astronomically oriented constructions. However, as the typological and functional diversity of civic and ceremonial architecture increased, the orientations previously embedded in E Groups were transferred to structures and groups of other types. Specific cases discussed below support this conclusion.

As determined on excavation maps kindly provided by Takeshi Inomata (September 2019), the orientation of the E Group of Ceibal, whose central axis was preserved along its various construction stages, was reproduced by both the Preclassic substructures and the Late Classic version of Structure A-24 [81]. The same orientation is materialized in Structure A-3, which was built in the Terminal Classic, but also has earlier stages [82, 83] (note that, due to secular variations in orbital elements mentioned in S1 Text, the dates delimiting an interval of 143 days tended to be March 2 and October 10 during the Preclassic, but moved to March 3 and October 11 during the Classic, when Structure A-3 was built; see S1 Table). By that time, the E Group lost its original significance and symbolism [44] and probably also its observational function, as indicated by the altered orientation of the buildings and the fact that the east-west axis was preserved, but no longer recorded the same sunrise dates because the increased height of the central structure on the eastern platform blocked the view to the natural horizon.

The E Group of Caracol, whose early stages date to the Late Preclassic, was oriented to sunrises on February 9 and November 1. During the Late Classic, the Central Acropolis was built east of the E Group. Since the central and highest building of the acropolis (Structure A37) was placed exactly along the east-west axis of the E Group, it could have served as a foresight for the observer standing on the E Group's western pyramid, which reached its current height during the same period [5, 32, 84, 85]. Incidentally, the orientations marking February 9 and November 1 (with an intervening interval of 100 days) occur elsewhere in Mesoamerica, including the Maya area [18, 20, 23], but the E Group of Caracol is one of the earliest known examples.

At Nixtun-Ch'ich', characterized by its gridded urban pattern, at least two E Groups share the same orientation and their central axes coincide with the main east-west axis of the site. Since the E Groups seem to be the earliest, Middle Preclassic constructions, they likely conditioned the development of the whole urban layout: along the same axis, other buildings and architectural groups with the same orientation were added later, including the massive Late Preclassic triadic group BB1 (Fig 10) [86–89]. The site axis was now clearly marked by the summits of these structures, which blocked the view to the horizon from E Groups, and may

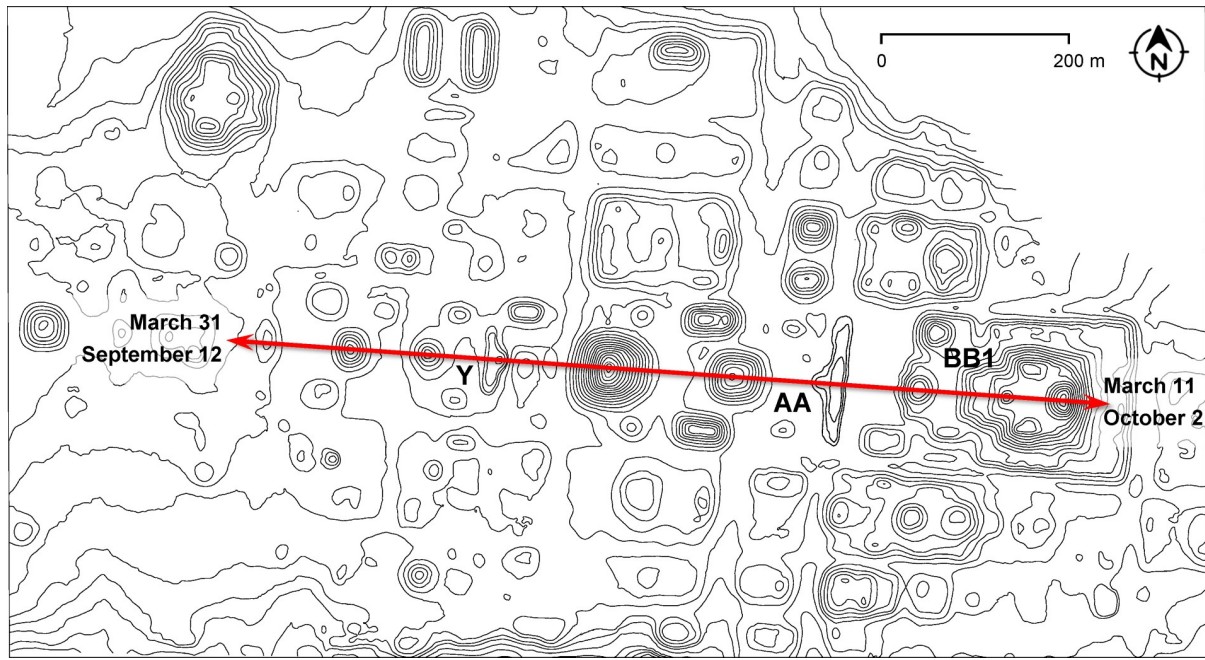

**Fig 10. Map of Nixtun-Ch'ich' (redrawn from [89]), with the main east-west site axis running over E Groups Y and AA and the triadic acropolis BB1.**

have been observationally functional in both directions; the dates recorded could have been integrated in a single observational scheme, in which the interval of 200 days was subdivided by multiples of 20 days (Fig 11). The alignment pertains to the most widespread orientation group in the central lowlands (Figs 5 and 6) and, contrary to what has been suggested [87, 89], cannot be related to the equinoxes.

A similar layout characterizes the core area of Cival. The main, Middle Preclassic E Group is skewed slightly north of east. Rather than the equinoxes [9], sunrises on the quarter-days of

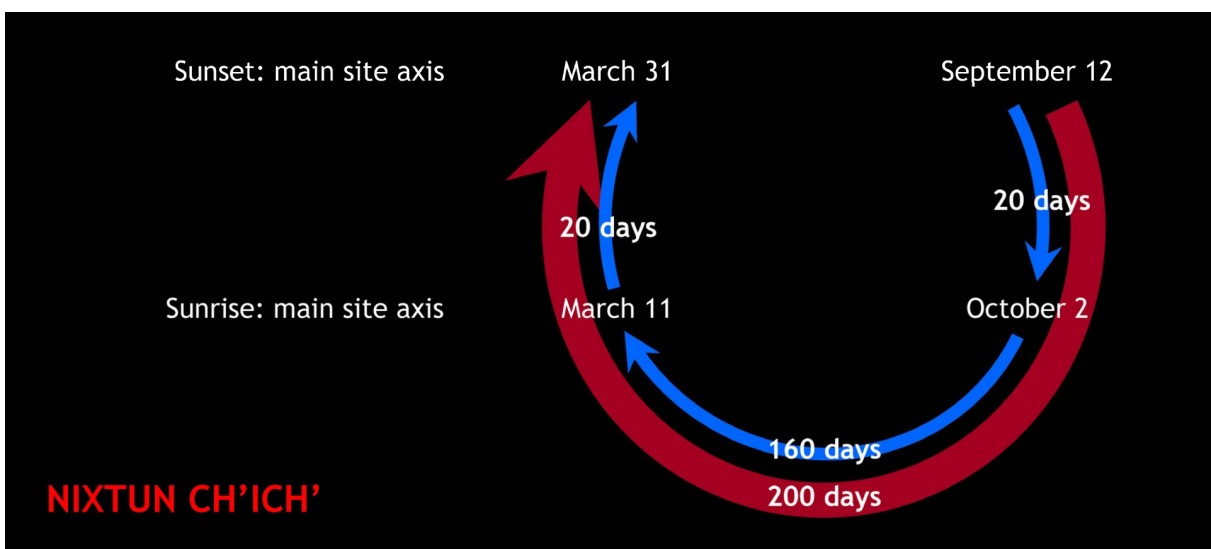

**Fig 11. Possible observational calendar of Nixtun-Ch'ich'.**

the year were a much more likely target of its central axis, which seems to have dictated the orientations of a number of surrounding structures and compounds, but became nonfunctional in the Late Preclassic, when the view to the eastern horizon was blocked by Group I, a massive triadic acropolis [9, 46], to which the observational function may have been transferred. Sunrises on the quarter-days were also marked by the E Group at Naranjo, whose initial versions date to the Late Preclassic and where the same orientation was materialized in the Late Classic Structure B-15, the most imposing building of the Central Acropolis [90–93]. This evidence reflects the importance of quarter-days in the area where a predilection for orientations deviated counterclockwise from cardinal directions is also attested (Fig 1).

At Tikal, the orientation of the Lost World E Group, whose central axis was established in the Middle Preclassic and maintained throughout its construction history, was reproduced by the adjacent Structure 5C-49 and also by Group H in the North Zone, both erected during the Classic period [42, 94, 95]. Structure 5D-46 of the Central Acropolis has a slightly different orientation, but it is noteworthy that this building, facing west, matches sunsets on March 31 and September 12. Since the E Group marked sunrises on March 11 and October 2, both pairs of dates could have composed a single observational calendar, identical to the one reconstructed for Nixtun-Ch'ich' (Fig 11). Such a functional relationship of both orientations is unlikely to be coincidental, considering that Structure 5D-46 was a residential palace of Chak Tok Ich'aak I, who is probably the occupant of the tomb excavated in the central building of the east platform of the E Group, where Stela 39 was also found, with an inscription commemorating the period ending ceremonies performed by this ruler in 376 CE [42, 96].

At Yaxhá, the orientation of the E Group in Plaza F is replicated by Plaza E and the Northeast Acropolis, with some buildings placed exactly along its central axis (Fig 12). Both Plaza F and the highest pyramid of the Northeast Acropolis, located at the eastern extreme of the alignment, were built in the Middle Preclassic or initial phases of the Late Preclassic period (Bernard Hermes, personal comm., July 2019). If observations were made from the western pyramid of the E Group, it was the pyramid on the Northeast Acropolis, rather than the much lower central mound of the eastern platform of the E Group, that would have served as a foresight, marking sunrises on March 2 and October 10, separated by 143 days. This interval must have been important at Yaxhá, because it also separates the sunsets on April 11 and September 1, most likely targeted by the North Acropolis, Plaza A, and the E Group of Plaza C, whose western directionality is supported by the—observationally unfavorable—proximity of the eastern horizon, formed by an elevation less than 300 m away, as well as by similar heights of the western pyramid and the central mound on the eastern platform. The four dates could have been incorporated in a single observational scheme composed of calendrically significant intervals (Fig 13). While Plaza C and North Acropolis date to the Late Preclassic, with possible earlier phases, Plaza A with the twin pyramids was built in the Late Classic [97].

At Naachtún, the North Plaza E Group has been dated to the transition from the Late Preclassic to the Early Classic period. Its central axis extended eastward passes over two mounds located about 500 m away and corresponds to sunrises on March 14 and September 30 (Fig 14). The slightly later triadic Structure I has the same orientation; its east-west axis of symmetry prolonged eastward passes over the Late Classic Structure XXXVIII, located over a kilometer away. It is thus highly likely that this pyramidal temple, which had its stairway on the western side and may have been oriented to Structure I, was deliberately built along the latter's axis. It is noteworthy that during the Late Classic, when the eastern sector of Naachtún became the main focus of the settlement, the western and earliest section became a sacred and funerary place, and a ceremonial path leading eastward from Structure I seems to have connected it with the eastern sector. The interval of 200 days marked by the above mentioned alignments on the eastern horizon must have been particularly important at Naachtún, as it also separated

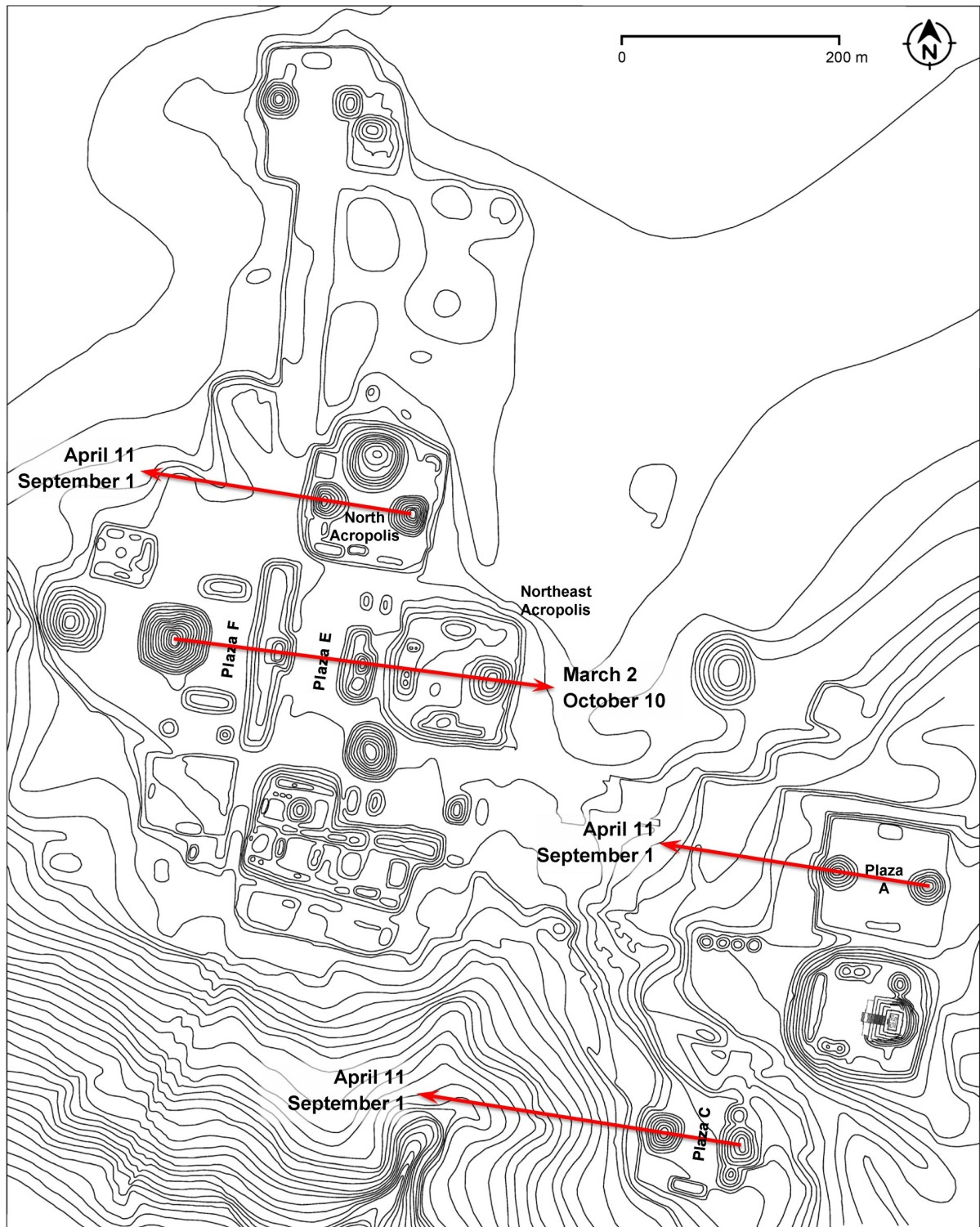

**Fig 12. Map of Yaxhá (redrawn from [98]), with alignments discussed in the text.**

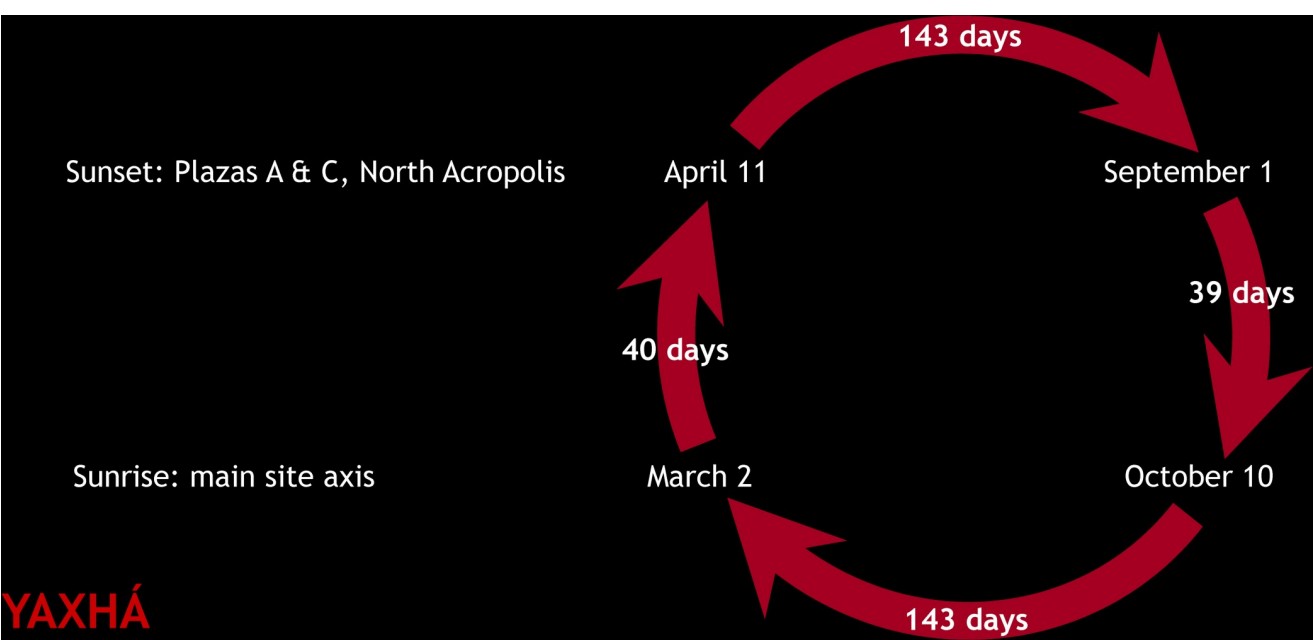

**Fig 13. Possible observational calendar of Yaxhá.**

sunsets on March 31 and September 12, corresponding to the orientation of the Late-Terminal Classic Structure XXXIX (Fig 14), which likely had its access from the west [99–105].

The cases discussed above illustrate the processes that resulted in similar orientations of E Groups and buildings of other types. Some differences that can nonetheless be observed between the distributions of dates most commonly recorded by E Groups (central axes) and other buildings (Fig 5) can be attributed to the fact that most E Groups were built in the Preclassic, while other structures are largely from later periods. In some regions E Groups continued to be in vogue until the Late Classic, e.g. in southeastern Petén [106], but their construction largely ceased in the Early Classic period [6].

Whether the shifts in the most frequently recorded dates reflect different strategies in agricultural scheduling, which may have been required as a consequence of climatic changes and variations in the length of the rainy season [76, 107, 108], is a question that cannot be answered with the data at hand, because the choice of relevant dates was likely affected by unknown variables, including the variety of maize cultivated, local environmental peculiarities, and idiosyncratic aspects of worldview and political ideology. The latter appear to have had a prominent role, promoting both tradition and innovation. On the one hand, the dates most commonly targeted by early constructions continued to be recorded throughout the Maya history, even if the frequency changed. On the other, some variations in orientation practices can best be explained in terms of local concepts and the autonomy of political entities [14]. The most evident case is the prevalent north-of-east/south-of-west skew of orientations in eastern Petén and western Belize (Fig 1): since the dates targeted were the same as elsewhere, though on the opposite horizons, this regional peculiarity cannot be accounted for by environmental constraints.

## E Groups: Astronomical observatories?

The results of the present study show that Group E-type complexes were not merely allegorical allusions to celestial cycles, without any observational function. Their orientations, indicated

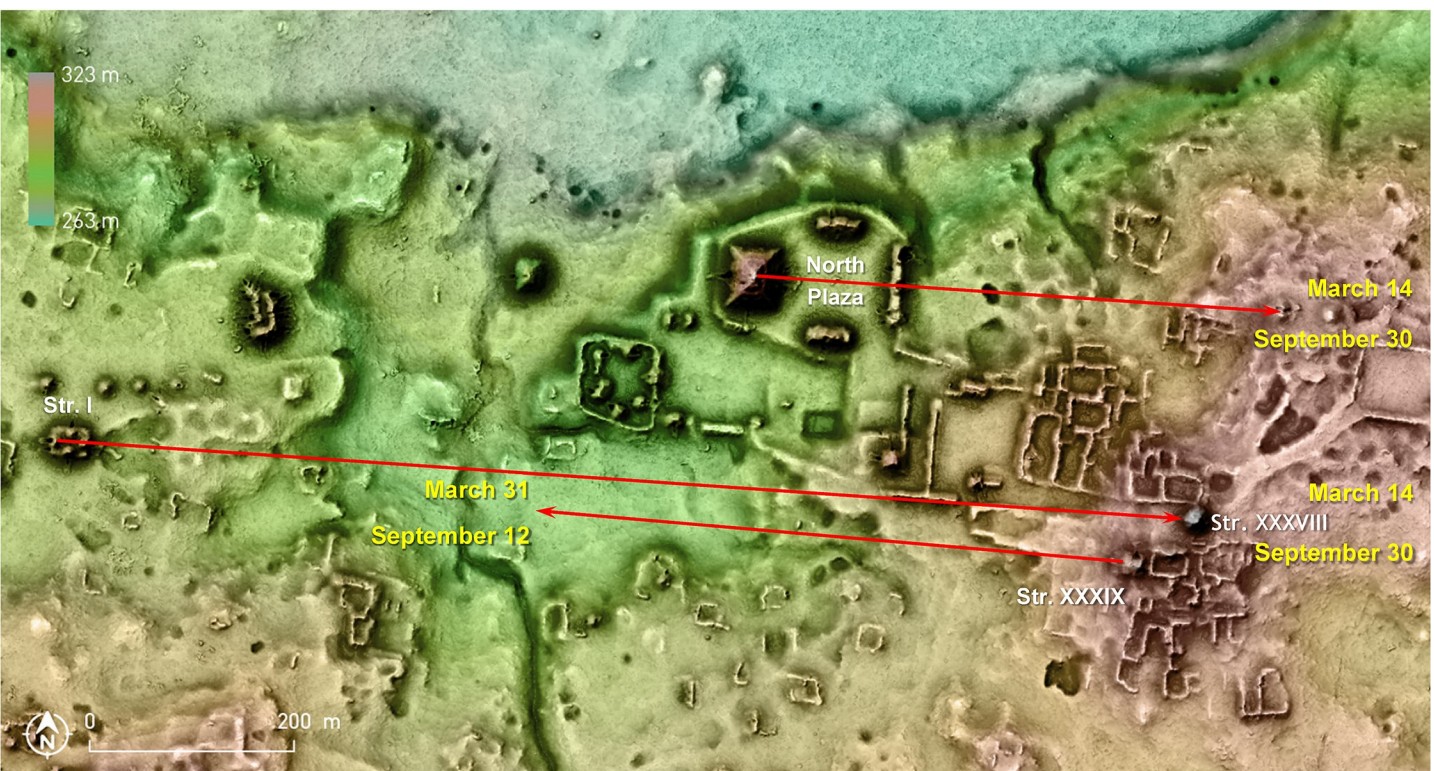

**Fig 14. Lidar-derived relief of Naachtún (extracted from the Pacunam PLI survey and provided by Philippe Nondédéo), with alignments discussed in the text (visualization by Žiga Kokalj).**

by their central axes, belong to widespread alignment groups, materialized mostly in buildings of other types and explicable in astronomical terms. Furthermore, the importance of the central axes is reflected in their longevity, the placement of offerings and burials, and the location and orientation of other buildings.

On the other hand, there is no archaeological evidence suggesting the importance of lateral alignments; as already mentioned, ritual deposits are sometimes found in the buildings on the extremes of eastern platforms, but they are not arranged along the putative astronomically significant visual lines. While it is not impossible that some lateral alignments were astronomically motivated, their general distribution offers no support to the idea that they were systematically used for sighting celestial events. Consequently, there are no reasons to suppose that E Groups were intended for particularly sophisticated astronomical observations, and most certainly not for tracking the equinoxes and the solstices. While an E Group in the narrowest sense is composed of the western pyramid and the eastern platform, topped by one or more buildings, it frequently integrates other buildings to the north and south, which commonly share the same orientation, indicated by the central east-west axis of symmetry of the compound. If only the latter was laid out on astronomical grounds, the observational functions of E Groups were not essentially different from those of other astronomically oriented buildings and complexes. This conclusion is supported by the fact that, while E Groups are concentrated in the central part of the Yucatan peninsula, the orientation groups they pertain to are spread all over the Maya Lowlands. Furthermore, chronological relationships known for several sites discussed above indicate that, while an E Group initially served for observations, this role was, at some point, transferred to other buildings with the same orientation. This is

particularly evident at the sites where these structures were placed along the central axis of the E Group; since they preserved the same orientation, but blocked the view to the horizon from the E Group, they were clearly intended to record the same celestial phenomena, which could no longer be observed from the E Group. The observational use of E Groups seems to have been often replaced by triadic complexes, which became popular in the Late Preclassic and were typically placed east of an E Group (Figs 10 and 12) [8, 109]. Finally, it should be considered that the orientations of some E Groups have no readily apparent astronomical correlates; it is highly likely that astronomical criteria did not dictate the orientation of each and every E Group, as is also true for other types of monumental buildings.

Comparing the orientations of E Groups with those of other contemporary (Late Preclassic and Early Classic) buildings in the Maya area, Aveni et al. [61] contended that E Groups followed special orientation criteria. However, most of E Groups they analyzed are in the central Yucatan lowlands, whereas the other orientations are from a broader area: those contributing to the concentration around solstitial azimuths—the most striking difference in comparison with orientations of E Groups—are largely from the Pacific coast and the adjacent highlands, where solstitial orientations prevailed in the Preclassic [14, 73]. Furthermore, as already pointed out [18], their comparison involves the north-south azimuths of eastern platforms of E Groups, which are not always exactly perpendicular to the east-west axes, and there are discrepancies between the histogram showing their distribution and the tabulated data.

Aveni et al. [61] also suggested that, "if seeing the sun can be shown to have been a part of the scheme, then regardless of whether the Maya were watching it scientifically or ceremonially, the associated architectural complex may be regarded as an observatory." There are reasons to disagree with this proposal. On the one hand, it is obvious that E Groups had multiple functions [4, 7–11, 109]; by qualifying them as observatories, we inevitably prioritize only one. On the other hand, astronomical orientations characterize the vast majority of civic and ceremonial buildings in the Maya area [12–15, 18, 19, 61, 73]; following the aforementioned opinion, all of them should be considered as observatories, which would be a simplification inconsistent with their typological diversity and functional complexity. While an "observatory", in the modern sense of the word, is a place to acquire knowledge, astronomically oriented Maya buildings represent, rather, the results of knowledge. Although they surely also served for monitoring the motion of celestial bodies, largely the Sun, this was not their primary use. In the light of these facts, the designation "astronomical observatory", applied to E Groups or any other type of Maya structures, is unwarranted and implies a biased and partial interpretation of their primarily ritual, residential, funerary or administrative functions.

## Conclusion

Special assemblages known as Group E-type complexes, particularly common in the central Yucatan peninsula, where they characterize almost all urban centers and even some minor settlements, were among the earliest Maya constructions with a formalized ground plan. While the details of their uses remain poorly understood, they evidently had in integrative role, promoting ritually sanctioned interaction and cooperation of the communities whose subsistence was based on different combinations of incipient agriculture and hunting-gathering strategies [79, 80]. Given their obvious socio-political significance and continued use through many centuries [110], a proper understanding of their possible astronomical functions, about which a number of different hypotheses have been proposed, is relevant to broader issues of Maya architectural and urban developments.

The above presented analyses of quantitative and contextual data have demonstrated that the orientations of E Groups belong to alignment groups explicable in astronomical terms and

largely materialized in buildings of other types throughout the Maya lowlands. Therefore, the opinion that they were observationally nonfunctional architectural allegories alluding to celestial phenomena cannot be sustained. On the other hand, the fact that only their east-west axes of symmetry were clearly based on astronomical criteria, whereas the sightlines from the western pyramid to the extremes of the eastern platform cannot be convincingly explained with astronomical motives, offers no support to the other, even more popular belief which associates E-Group assemblages with particularly sophisticated observational practices.

In the Maya area, like elsewhere in Mesoamerica, most of the important civic and ceremonial buildings were oriented to sunrises and sunsets on agriculturally significant dates, which tend to be separated by multiples of elementary periods of the Mesoamerican calendrical system. The distribution of dates in the year and contextual evidence, including ethnographic data, suggest that the solar alignments allowed the use of observational schemes that were easily manageable by means of the formal calendar, thus facilitating prediction of important dates in the seasonal cycle. Aside from constructing and managing irrigation or drained field systems intended to mitigate the risk posed by agricultural way of life [111, 112], an efficient regulation of farming activities in the yearly cycle must have been of vital importance, but could only be based on astronomical observations, since the 365-day calendrical year, due to the lack of intercalations, did not maintain a permanent concordance with the tropical year.

However, the astronomical alignments cannot be understood in purely utilitarian terms. Since the dates recorded by orientations were based on intervals easily manageable with the aid of the ritual 260-day cycle, they must have marked canonical moments appropriate for performing agriculturally important ceremonies. The rituals directed to supernatural forces and intended to secure a proper sequence of seasonal changes, the growth of cultigens, and abundance of crops were, obviously, no less important than an adequate management of agricultural labor. In Trigger's words [113], those who commissioned monumental constructions "would have viewed theological goals, such as serving and winning the favour of the gods, as being highly practical." Besides, the simple objective of timekeeping by means of solar observations could have been achieved without monumental constructions, even without archaeologically recoverable artifacts. The astronomically oriented buildings reified the beliefs about the structure and functioning of the universe. If the apparently perfect order observed in the sky, evidently superior to that on Earth, was the primary source of deification of heavenly bodies [13, 114], their cyclic behavior was not viewed as simply correlated with seasonal transformations in natural environment, but rather as provoking them. Consequently, if timely occurrences of these changes were believed to be conditioned by the arrival of the Sun to specific points on the horizon, the architectural alignments reproducing directions to these phenomena may well have been intended to ensure, in accordance with the principles of magic, their regular sequence.

The same agricultural and ritual concerns are reflected in E Groups. The orientations of the earliest E Groups were consistently replicated in later cases, clearly demonstrating their astronomical basis. Considering the use of orientations for scheduling of agricultural and related ceremonial activities, it is significant that the earliest E Groups appeared precisely at a time when farming was becoming a predominant subsistence strategy in the Maya Lowlands. With the increasing architectural diversity in the following centuries, the orientations originally embedded in E Groups, as well as their observational and ritual functions, were transferred to buildings and complexes of other types, thus substantially affecting the appearance of urban layouts. At several sites we can observe that the view to the horizon from an E Group was, at some point, blocked by higher constructions erected along its central axis and adopting the same orientation. This fact, as well as the longevity of central axes archaeologically demonstrated in several E Groups, clearly attests to the long-lasting significance of the initially

intended astronomical referents. The overall cultural development, including relative autonomy of political entities and the continuously changing political geography, resulted in some regional and time-dependent differences in orientation practices, but the persistence of the most widespread alignment groups up to the Spanish Conquest mirrors their practical and symbolic importance.

None of the currently known astronomically oriented structures can be interpreted as an observational device in the modern scientific sense. Since their primary functions were religious, residential, or administrative, the term "observatory" applied to either E Groups or buildings of other types is clearly inappropriate. The principles underlying Maya architecture and urban layouts cannot be comprehensively understood without considering the interdependence of "practical", subsistence-related issues, religious norms, and ritual observances, but the role of astronomically-derived concepts in this complex set of rules has been largely underappreciated. The orientations of both E Groups and other monumental constructions in civic and ceremonial cores of ancient settlements reflect the importance of these concepts in political ideology. An appropriate timing of agricultural tasks and ritual performances contributed to the legitimation of power of the ruling class and thus reinforced social cohesion necessary for preserving the existing political order. In the light of these arguments, it is precisely the importance of the astronomically and cosmologically significant directions that allows us to understand some prominent aspects of ancient Maya architecture and urbanism.

## Supporting information

**S1 Text. Details on materials and methods.**
(DOCX)

**S1 Table. Alignment data for E Groups and structures of other types.**
(XLSX)

## Acknowledgments

The alignment data for this study were acquired within the research project *Estudio Arqueoastronómico de la Arquitectura Maya en Petén* (2019), co-directed by Claudia Marie Vela González de Bellamy, authorized by the *Dirección del Patrimonio Cultural y Natural*, *Ministerio de Cultura y Deportes*, Guatemala (Convenio de Investigación Arqueológica núm. 17–2019) and additionally supported by Dieter Richter, and within the project *Propiedades astronómicas de la arquitectura y el urbanismo en Mesoamérica* (2010–2015), co-directed by Pedro Francisco Sánchez Nava and authorized by the *Instituto Nacional de Antropología e Historia* (INAH), Mexico. Lidar data used for determining alignments in several E Groups are not publicly available. Upon the author's request, limited portions of lidar imagery of Belize were made available by Arlen F. Chase and Adrian S. Z. Chase, to whom I am also indebted for important archaeological information. Structure data derived from the Pacunam's 2016 lidar survey in Guatemala were provided by Philippe Nondédéo (for the Naachtún area) and Francisco Estrada-Belli (for the area of Holmul and Cival). Takeshi Inomata shared the lidar-derived digital elevation model of the Ceibal area, as well as excavation maps of Ceibal. I wish to express my sincere thanks to all these colleagues and institutions.

## Author Contributions

**Conceptualization:** Ivan Šprajc.

**Data curation:** Ivan Šprajc.

**Formal analysis:** Ivan Šprajc.

**Funding acquisition:** Ivan Šprajc.

**Investigation:** Ivan Šprajc.

**Methodology:** Ivan Šprajc.

**Project administration:** Ivan Šprajc.

**Supervision:** Ivan Šprajc.

**Validation:** Ivan Šprajc.

**Writing – original draft:** Ivan Šprajc.

**Writing – review & editing:** Ivan Šprajc.

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
