## [Decision Letter · Decision Letter 0]

20 Nov 2020

PONE-D-20-26825

Astronomical aspects of Group E-type complexes and implications for understanding ancient Maya architecture and urban planning

PLOS ONE

Dear Dr. Šprajc,

Thank you for submitting your manuscript to PLOS ONE. After careful consideration, we feel that it has merit but does not fully meet PLOS ONE’s publication criteria as it currently stands. Therefore, we invite you to submit a revised version of the manuscript that addresses the points raised during the review process.

Your paper has been reviewed by two well accomplished scholars of Mesoamerican archaeology, including one with some expertise in archaeoastronomy. Reviewer 1 believes that the paper meets Plos One’s standards of scientific soundness. Reviewer 2 believes that the paper does not meet Plos One’s standards of scientific soundness.  Reviewer 2 raises important critiques that must be adequately addressed before the paper can meet Plos One’s publication criteria. I list the three most important points below (though all critiques from both reviewers are useful).

``The author has not demonstrated that the observed observations are not due to chance. There is some clustering of alignments with the (rather large) range of predicted values, but many alignments are far from a prediction.”  This bears directly on the criterion of appropriate and rigorous statistical analysis. In addition to R2’s critique, from my own reading, the justification of the use of KDEs needs more development.``Most of the exercise seems composed of post-hoc reasoning, without testing of hypotheses. How would one know if the author is wrong? No empirical means of choosing between a correct and an incorrect interpretation are given.” This bears directly on the issue of whether the data support the conclusions of the paper.More evidence is needed to justify the assertion that calendars were used to ``facilitate an efficient scheduling of agricultural works.”

We look forward to receiving your revised manuscript.

Kind regards,

Jacob Freeman

Academic Editor

PLOS ONE

Journal Requirements:

2. In your manuscript, please provide additional information regarding the specimens used in your study. Ensure that you have reported specimen numbers and complete repository information, including museum name and geographic location.

For more information on PLOS ONE's requirements for paleontology and archaeology research, see https://journals.plos.org/plosone/s/submission-guidelines#loc-paleontology-and-archaeology-research.

Reviewers' comments:

Reviewer's Responses to Questions

**Comments to the Author**

1. Is the manuscript technically sound, and do the data support the conclusions?

Reviewer #1: Yes

Reviewer #2: Partly

2. Has the statistical analysis been performed appropriately and rigorously? 

Reviewer #1: Yes

Reviewer #2: No

3. Have the authors made all data underlying the findings in their manuscript fully available?

Reviewer #1: Yes

Reviewer #2: Yes

4. Is the manuscript presented in an intelligible fashion and written in standard English?

Reviewer #1: Yes

Reviewer #2: Yes

5. Review Comments to the Author

Reviewer #1: This is a well-executed and well-written paper that deserves to be published. Sprajic has presented a solid contribution in archaeoastronomy in which he reanalyzes the function of E Groups, an architectural complex that forms the earliest public architecture at most ancient Maya sites. Whereas they were originally conceived to have reflected the passage of the sun in terms of solstices and equinoxes, Sprajic demonstrates that this cannot be the case and instead argues that they only share common east-west alignments that could have been useful in the past in breaking up the solar year into smaller units that were likely related to agricultural processes. His command of archaeoastronomy goes beyond most of his peers and he has done much to redeem this form of research by explicitly documenting his research processes. For anyone interested in the origins of Maya, and Mesoamerican, civilization, this article will be extremely useful.

detailed comments:

Line 75: besides “principles of Maya architectural and urban planning,” these alignments could be representative of Maya cosmological systems

Line 179: rather than “its,” the word should probably be “their”

Line 225: since Ceiba is part of Caracol in the Late Classic Period, perhaps “locale” is better than “site”

Line 325: need to clarify that lateral alignments does not refer to deposits in lateral buildings, which do occur at Uaxactun, Caracol, and Tikal during the Early Classic era

Line 371: should also cite reference [5] for the Early Classic date of this western pyramid

Line 504: please explain what this shift did

Line 648: add a reference for construction largely ceasing in the Early Classic Period (perhaps reference [6])

Line 669: the lateral buildings sometimes hold later deposits, but do not reflect the original alignments

Line 720: suggest adding new reference after “through many centuries”:

Chase, DZ, McAnany, PA, Sabloff, JA. Epilogue: E Groups and their Significance to the Ancient Maya. In: Freidel, DA, Chase, AF, Dowd, AS, Murdock, J, editors. Maya E Groups: Calendars, astronomy, and urbanism in the early lowlands. Gainesville: University Press of Florida; 2017, p. 578-582.

Reviewer #2: I find much that is problematic in this paper, but I am not sufficiently versed in the methods and theory of astronomy or archaeoastronomy to present a proper critique. The following are among the negative features that occur to me.

1. The author has not demonstrated that the observed observations are not due to chance. There is some clustering of alignments with the (rather large) range of predicted values, but many alignments are far from a prediction.

2. It seems implausible to me that the basic principle of orientation to alignments whose sunrise dates are separated by 20 days, or any of a series of multiples of 20 days. This is a basic claim of this author in many publications over the years, but it does not seem plausible to me. The author has proposed this not only for the Maya, but also for the Aztecs and other Mesoamerican cultures. I would think that if such a practice was indeed pursued in ancient Mesoamerica—particularly in the Aztec case—there would be some clue or hint of it in the copious ethnohistoric literature on the Aztecs. The closest reference is a single documentary account that the Aztec main temple was aligned with sunrise on the solstice, which does not mention 20-day intervals or multiples thereof.

3. Most of the exercise seems composed of post-hoc reasoning, without testing of hypotheses. How would one know if the author is wrong? No empirical means of choosing between a correct and an incorrect interpretation are given.

4. Many of the author’s claims, including some very unlikely claims, are justified by citations to the author’s own works, with little or no consideration of the work of other scholars. The level of self-citation is remarkably high in this paper. Wild, implausible ideas are justified with self-citation, which only casts doubt on the use of self-citation to justify ideas that seem less problematic.

The author expresses the idea that Mesoamerican observational calendars were “intended to facilitate an efficient scheduling of agricultural works” (p. 21, line 479), lists four obscure publications of his own as justification, saying they contain the details (see also p. 31, lines 737-741). This is a hold-over from the long-discredited view that Maya priests ruled society, deriving their power form their control of the calendar. Without priests interpreting a calendar, peasants would not know when to plant or harvest. This is simply nonsense. Farmers know when to plant and harvest from environmental cues; there is ample ethnographic confirmation of this from small-scale peasant farming today. I have read many ethnographies that confirm this common-sense viewpoint, and I have talked with many Mesoamerican peasants about how they schedule major activities like planting and harvesting. Calendars have little to do with this, apart perhaps from defining the start of a period when farmers begin following environmental cues more closely. The key variable for timing the planting (in central Mexico, at least) is the onset of the spring rains, and this does not happen on a set date. Calendars were NOT needed for “efficient scheduling of agricultural works.” This claim is repeated—even amplified—on page 31, lines 737-741), where we are told that efficient agriculture “could only be based on astronomical observations.” Again, this is nonsense.

The section on E-groups at Maya sites (starting p. 22) is full of speculation, much of it somewhat wild and unconstrained (lines 520-521, or 531-532). Also, the conclusions are full of speculations.

I don’t think any specialists would agree with the assertion (p. 30, line 703) that “astronomical orientations characterize the vast majority of civic and ceremonial buildings in the Maya area” (with self-citations as justification). I have discussed this issue with a number of Maya archaeologists, and none of them would agree with this assertion. Their view is that the astronomical work of Sprajc and others is a fringe activity that can be ignored for its improbable interpretations. I must admit that I am not aware of published critiques of this assertion. It is repeated on page 31 (lines 732).

I don’t understand figures 11 and 13.

6. PLOS authors have the option to publish the peer review history of their article (what does this mean?). If published, this will include your full peer review and any attached files.

Reviewer #1: No

Reviewer #2: No

---

## [Author Response · Author response to Decision Letter 0]

8 Dec 2020

My responses to the editor and reviewers are given in the attached file Response to Reviewers.docx

---

## [Editor Report · Decision Letter 1]

14 Apr 2021

Astronomical aspects of Group E-type complexes and implications for understanding ancient Maya architecture and urban planning

PONE-D-20-26825R1

Dear Dr. Šprajc,

We’re pleased to inform you that your manuscript has been judged scientifically suitable for publication and will be formally accepted for publication once it meets all outstanding technical requirements.

Kind regards,

Jacob Freeman

Academic Editor

PLOS ONE

Additional Editor Comments (optional):

I apologize for the delayed decision. It took some time to do due diligence. Thank you for submitting your paper to Plos One and for bearing with a difficult and long review process.
---

## [Editor Report · Acceptance letter]

16 Apr 2021

PONE-D-20-26825R1 

Astronomical aspects of Group E-type complexes and implications for understanding ancient Maya architecture and urban planning 

Dear Dr. Šprajc:

I'm pleased to inform you that your manuscript has been deemed suitable for publication in PLOS ONE. Congratulations! Your manuscript is now with our production department. 

Kind regards, 

on behalf of

Dr. Jacob Freeman 

Academic Editor

PLOS ONE